# A fungal RNA-dependent RNA polymerase is a novel player in plant infection and cross-kingdom RNA interference

An-Po Cheng[1], Bernhard Lederer[1�], Lorenz Oberkofler[1�], Lihong Huang[1¤], Nathan R. Johnson[2,3], Fabian Platten[1], Florian Dunker[1], Constance Tisserant[1], Arne Weiberg[1]*

**1** Institute of Genetics, Faculty of Biology, Ludwig Maximilian University of Munich, Martinsried, Germany, **2** Centro de Genómica y Bioinformática, Facultad de Ciencias, Ingeniería y Tecnología, Universidad Mayor, Santiago, Chile, **3** ANID-Millennium Science Initiative Program-Millennium Institute for Integrative Biology (iBio), Santiago, Chile

☯ These authors contributed equally to this work.
¤ Current address: GMU-GIBH Joint School of Life Sciences, The Guangdong-Hong Kong-Macau Joint Laboratory for Cell Fate Regulation and Diseases, Guangzhou Medical University, Guangzhou, China
* a.weiberg@lmu.de

**Data Availability Statement:** Sequencing data have been deposited in NCBI SRA (BioProject ID PRJNA978613).

## Abstract

Small RNAs act as fungal pathogen effectors that silence host target genes to promote infection, a virulence mechanism termed cross-kingdom RNA interference (RNAi). The essential pathogen factors of cross-kingdom small RNA production are largely unknown. We here characterized the RNA-dependent RNA polymerase (RDR)1 in the fungal plant pathogen *Botrytis cinerea* that is required for pathogenicity and cross-kingdom RNAi. *B. cinerea bcrdr1* knockout (ko) mutants exhibited reduced pathogenicity and loss of cross-kingdom small RNAs. We developed a "switch-on" GFP reporter to study cross-kingdom RNAi in real-time within the living plant tissue which highlighted that *bcrdr1* ko mutants were compromised in cross-kingdom RNAi. Moreover, blocking seven pathogen cross-kingdom small RNAs by expressing a short-tandem target mimic RNA in transgenic *Arabidopsis thaliana* led to reduced infection levels of the fungal pathogen *B. cinerea* and the oomycete pathogen *Hyaloperonospora arabidopsidis*. These results demonstrate that cross-kingdom RNAi is significant to promote host infection and making pathogen small RNAs an effective target for crop protection.

## Author summary

*Botrytis cinerea* is a notorious plant pathogen that can only be effectively controlled by chemical fungicides. With the aim to reduce fungicide application, new control strategies are in need. Cross-kingdom RNA interference (RNAi) is an emerging field in plant-pathogen research. Uncovering the key factors to understand the molecular mechanisms and functions in cross-kingdom RNAi is fundamental to develop innovative RNA-based strategies for crop protection. *B. cinerea* produces extracellular small RNAs to induce natural cross-kingdom RNAi in plant hosts for infection. In this study, we describe the *B. cinerea*

**Funding:** AW received funding for this study from the German Research Foundation (DFG, www.dfg.de), grant award no. WE5707/2-1 as part of the Research Unit FOR5116. The funder did not play any role in the study design, data collection and analysis, decision to publish, or presentation of the manuscript.

**Competing interests:** The authors have declared that no competing interests exist.

BcRDR1 as a novel pathogenicity factor that is required for small RNA production and cross-kingdom RNAi. Establishing an advanced GFP-based switch-on reporter expressed transgenic plants allowed us to visualize and record the dynamics of cross-kingdom RNAi during infection. We transformed this newly acquired information to design pathogen small RNA-specific target mimics, so-called short tandem target mimics, to express in transgenic plants, which led to reduced disease severity. Our study not only expanded the knowledge of the molecular functions in cross-kingdom RNAi, but also showcased how such gained knowledge can be instrumental to interfere with pathogen cross-kingdom RNAi for a better pathogen control.

## Introduction

RNA-dependent RNA polymerases (RdRPs, RDRs) synthesize a complementary RNA strand from a primary RNA template without requiring any free DNA or RNA primer. Two classes of RdRPs/RDRs have been described. The viral-type RdRPs have been well-characterized in viral replication and have been also found as parts of retrotransposon genomes. Viral-type RdRPs are promising therapeutic targets to cure COVID-19 and other viral infections [1]. The cellular-type RDRs are conserved in several eukaryote kingdoms, including plants, animals, and fungi, and are functional in RNA interference (RNAi) pathways [2,3]. RDRs generate double-stranded RNA precursors for the DICER or Dicer-like (DCL)-dependent biogenesis of primary and secondary small-interfering (si)RNAs that load into Argonaute (AGO) proteins and form an RNA-induced silencing complex (RISC) inducing transcriptional or post-transcriptional gene silencing.

In the model plant *Arabidopsis thaliana*, the RDR2 synthesizes from DNA polymerase IV-dependent RNA transcripts double-stranded RNAs (dsRNAs) that initiate small RNA-directed DNA methylation (RdDM). RDR2 and the RdDM pathway are mainly responsible for the transcriptional control of transposons and are involved in transgene-induced gene silencing [4]. The *A. thaliana* RDR6 is required to produce secondary siRNAs, an amplification loop that fosters gene silencing of RNA virus and transgenes, as well as controlling endogenous gene expression. Plant RDR2 and RDR6 are both involved in the molecular stress response and defense against various types of plant pathogens, including virus, bacteria, fungi, and oomycetes [5,6].

RDRs are also required for RNAi and the biogenesis of various types of small RNAs in fungi [7]. Two distinct RDR-dependent RNAi pathways have been elucidated in the fungus *Neurospora crassa*. The RDR QDE-1 is part of the quelling pathway, an RNAi mechanism that controls transgene-induced gene silencing. A second *N. crassa* RDR, SAD-1, produces dsRNAs to induce meiotic silencing of unpaired DNA (MSUD), controlling aberrant RNA accumulation during late meiosis [2]. Two distinct RDRs of the basal fungus *Mucor circinelloides* are required for initiation and amplification of transgene-induced gene silencing [8]. Like plants, fungal RDRs in the species *Cryphonectria parasitica* are functional in antiviral defense and transposon silencing [9]; however, a RdDM pathway as known in plants has not been found in fungi. Moreover, RDRs have been proposed to regulate genes affecting fungal growth and development, but little is known about the functional role of fungal RDRs in pathogenicity. A knockout of *MoRdRP1* in the rice blast fungus *Magnaporthe oryzae* led to reduced growth, conidia formation, and disease symptoms on rice leaves [10]. Similarly, *fgrdrp2*, *fgrdrp3* and *fgrdrp4* knockout strains of the head blight inducing fungal pathogen *Fusarium graminearum* indicated reduced conidia, mycotoxin production, and head blight on inoculated wheat spikes

[11]. While such reports have indicated that distinct fungal RDRs are involved in pathogenicity, their mode of action remains unclear.

In this study, we investigated the role of the *BcRDR1* gene in the fungus *Botrytis cinerea*, a multi-host plant pathogen infecting more than 1,000 different plant species and causing the devastating grey mold disease. As part of its pathogenicity, *B. cinerea* releases extracellular small RNAs (Bc-sRNAs) that enter plant cells during infection and recruit the plant's own AGO1/RISC to induce host gene silencing for promoting infection [12,13]. This virulence phenomenon has been termed cross-kingdom RNAi and is a general infection strategy of diverse plant and animal pathogenic as well as symbiotic microbes, including fungi, oomycetes, and bacteria [14–16]. Most Bc-sRNAs inducing cross-kingdom RNAi are encoded by GYPSY-class of long terminal repeat retrotransposons and are produced by *B. cinerea* BcDCL1 and BcDCL2 [13,15,17]. We herein discovered that the *B. cinerea* BcRDR1 is a pathogenicity factor and is required for *B. cinerea*-induced cross-kingdom RNAi.

## Results

### *B. cinerea* BcRDR1 is a pathogenicity factor

In order to identify unknown genes involved in cross-kingdom RNAi, we analyzed the family of BcRDRs in *B. cinerea*. Through a homology search with the *N. crassa* RDR, SAD-1, we identified three genes encoding conserved RDRs in the genome of the *B. cinerea* strain B05.10, thereafter named *BcRDR1* (*Bcin01g01810*), *BcRDR2* (*Bcin08g02220*), and *BcRDR3* (*Bcin07g01750*). Phylogenetic analysis of full-length amino acid sequences revealed that BcRDR1 is the orthologue of the *N. crassa* SAD-1 involved in MSUD [18], and BcRDR2 is the orthologue of *N. crassa* quelling RDR, QDE-1 [19]. BcRDR3 is another conserved RDR with unknown function (S1A Fig). All three BcRDRs comprised a conserved DxDGD motif indicating that they function as RNA polymerases (S1B Fig). We detected transcripts of *BcRDR1*, *BcRDR2* and *BcRDR3* when growing *B. cinerea* in liquid culture or during infection of *Solanum lycopersicum* (tomato) leaves; however, there was no consistent up-regulation of any of the *BcRDR*s measured during infection (S2 Fig).

In order to investigate potential functions of BcRDRs in *B. cinerea* pathogenicity, we generated targeted gene knockout (ko) mutant strains by homologous recombination. We isolated two independent *bcrdr1* ko and two independent *bcrdr2* ko isolates (S3 Fig), while we obtained only heterozygous isolates when attempting to generate *bcrdr3* ko mutants. Therefore, we continued analyzing *bcrdr1* and *bcrdr2* ko mutants. Infection assays using detached *S. lycopersicum* or *A. thaliana* leaves revealed that *bcrdr1* ko mutants induced smaller lesion sizes compared to the *B. cinerea* wild type (WT) strain (Fig 1A and 1B). This phenotype was accompanied with lower fungal biomass, as estimated by quantification of *B. cinerea* genomic DNA, while genetic complemented BcRDR1 (cBcRDR1) strains in the *bcrdr1* ko background were reverted to full virulence (Figs 1C and S4). Both *bcrdr1* ko mutants and cBcRDR1 strains showed a WT-like growth and development phenotype under axenic culture condition (Figs 1A and S5). The *bcrdr1* ko mutants were not affected in the expression of other known virulence genes [20] (S6 Fig). Based on these results, we concluded that BcRDR1 is a pathogenicity factor in *B. cinerea* which was likely related to Bc-sRNAs.

### Biogenesis of retrotransposon-derived Bc-sRNAs requires BcRDR1

Some plant, nematode, and fungal RDRs are required for siRNA biogenesis, and abolished Bc-sRNA production in a *B. cinerea* *bcdcl1dcl2* ko mutant led to reduced pathogenicity due to impaired cross-kingdom RNAi [13]. To investigate the role of BcRDR1 in Bc-sRNA biogenesis, we performed comparative small RNA-seq analysis. *B. cinerea* WT and two independent

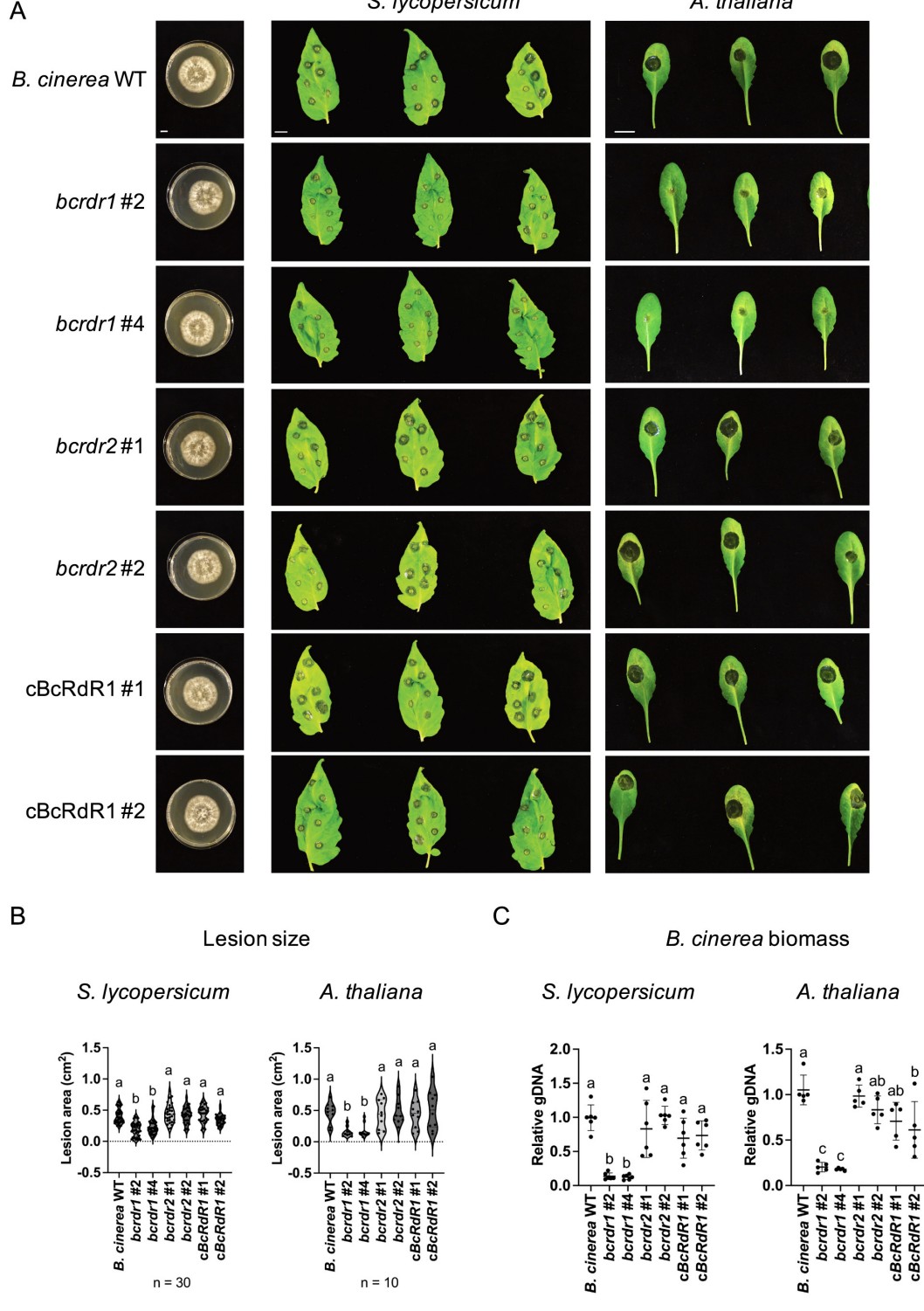

**Fig 1. *B. cinerea bcrdr1* ko mutants are compromised in pathogenicity. A)** Infection series on *S. lycopersicum* or *A. thaliana* detached leaves using *B. cinerea* WT, two *bcrdr1* ko mutants #2 and #4, two *bcrdr2* ko mutants #1 and #2, as well as two cBcRDR1 complementation strains #1 and #2. For each strain, a 20 µl drop of 5 x 10^4/ml or 2 x 10^5/ml conidiospore suspension was placed onto *S. lycopersicum* or *A. thaliana* leaves, respectively. **B)** Lesion size induced by *B. cinerea* infection was measured at 48 hpi for *S. lycopersicum* or 60 hpi for *A. thaliana*. 30 lesions on *S. lycopersicum* and 10 lesions on *A. thaliana* were

measured and statistical analysis was performed using ANOVA followed by a Tukey post-hoc test with *p*-value threshold *p* < 0.05. The scale bars represent 1 cm. C) *B. cinerea* biomass was estimated by measuring genomic DNA using primers of the *Bc-Tubulin (BcTub)A* and related to plant genomic DNA measured with *SlActin2* or *AtActin2* primers in five or six biological replicates. Statistical analysis was performed using ANOVA followed by a Tukey post-hoc test with *p*-value threshold *p* < 0.05. The scale bars represent 1 cm.

*bcrdr1* ko mutants were cultured for two days in liquid medium, and mycelium was collected to extract total RNA. Upon small RNA isolation via PAGE, RNA libraries were cloned for Illumina sequencing. In total, sequencing depth among the small RNA libraries were in the range of 1.3–4.9 millions that were mapped to the *B. cinerea* reference genome either unique or multiple times, as to the same read mapped to multiple chromosomal regions. The fraction of multiple mapping reads was reduced in both *bcrdr1* ko mutants (Fig 2A). Plotting read numbers by size revealed a reduction of 21–22 nucleotides (nt) long Bc-sRNAs in both *bcrdr1* ko mutants, mostly representing 5'prime Uracil nucleobase, retrotransposon (RT)-derived Bc-sRNAs (Figs 2B–2D and S7). This result indicated that production of Bc-sRNAs that mostly mapped to RTs was dependent on BcRDR1. The long-terminal repeat RT *BcGypsy3* was previously found to be a major source of Bc-sRNAs that were induced during plant infection and was a pathogenicity factor of *B. cinerea* [17]. Accordingly, reduced accumulation of RT-derived Bc-sRNAs in *bcrdr*1 ko mutants resulted in increased mRNA levels of *BcGypsy3* (S7C Fig) and upregulation of BcGypsy mRNAs during tomato infection was only measured with *B. cinerea* WT, but not with *bcrdr1* ko mutants (S7D–S7E Fig), suggesting a potential role of BcRDR1 in post-transcriptional silencing of RTs.

## *B. cinerea bcrdr1* ko mutants are compromised in cross-kingdom RNAi

RT-derived Bc-sRNAs were previously found to induce cross-kingdom RNAi [13,17]. In particular, the four Bc-sRNAs, Bc-sRNA3.1, Bc-sRNA3.2, Bc-sRNA5, and Bc-sRNA20 induced silencing of the *S. lycopersicum* host genes *Vacuolar sorting protein* (*SlVPS*) (*Solyc09g014790*), *Mitogen-activated protein kinase kinase kinase* (*SlMPKKK)4* (*Solyc08g081210*), and *C2H2 zinc-finger transcription factor SlBhlh63* (*Solyc03g120530*), and the *A. thaliana* host genes *Mitogen-activated protein kinase* (*AtMPK)1* (*AT1G10210*), *AtMPK2* (*AT1G59580*), a *Cell Wall kinase* (*WAK*) (*AT5G50290*), and the *Peroxiredoxin PRXIIF* (*AT3G06050*) (Fig 3A). We confirmed that *bcrdr1* ko mutants lost accumulation of these Bc-sRNAs, while they were produced in the *B. cinerea* WT and the cBcRDR1 strains during tomato infection as revealed by stem-loop reverse transcriptase PCR (Figs 3B and S8). When inoculating *S. lycopersicum* or *A. thaliana* with *B. cinerea* WT or *bcrdr1* ko mutants, we observed significant down-regulation of Bc-sRNA target genes in WT-infected plants compared to non-inoculated plants. The target gene down-regulation was abolished or less strong in *bcrdr1*-infected plants (Figs 3C and S9). Successful infection of plants and induced host defense response due to *B. cinerea* infection was validated by measured up-regulation of *S. lycopersicum Proteinase inhibitor* (*SlPI*)-*I* and *SlPI-II* or *A. thaliana Plant Defensin* (*AtPDF)1.2* immunity genes (Fig 3D). These results indicated that *bcrdr1* mutants might be compromised in inducing cross-kingdom RNAi, which would explain the reduced pathogenicity observed in *S. lycopersicum* and *A. thaliana*.

To further inspect whether *bcrdr1* mutant strains were indeed compromised in inducing cross-kingdom RNAi, we designed a GFP "switch-on" cross-kingdom RNAi reporter that exclusively responded to the translocation of Bc-sRNA3.1 and Bc-sRNA3.2 from *B. cinerea* into the plant host (Fig 4A). In this reporter construct, the CRISPR-type RNA endonuclease Csy4 [21] is co-expressed with a GFP version that is fused to the Csy4 recognition motif at its N-terminus, in adaptation to a previously designed cross-kingdom RNAi reporter for the

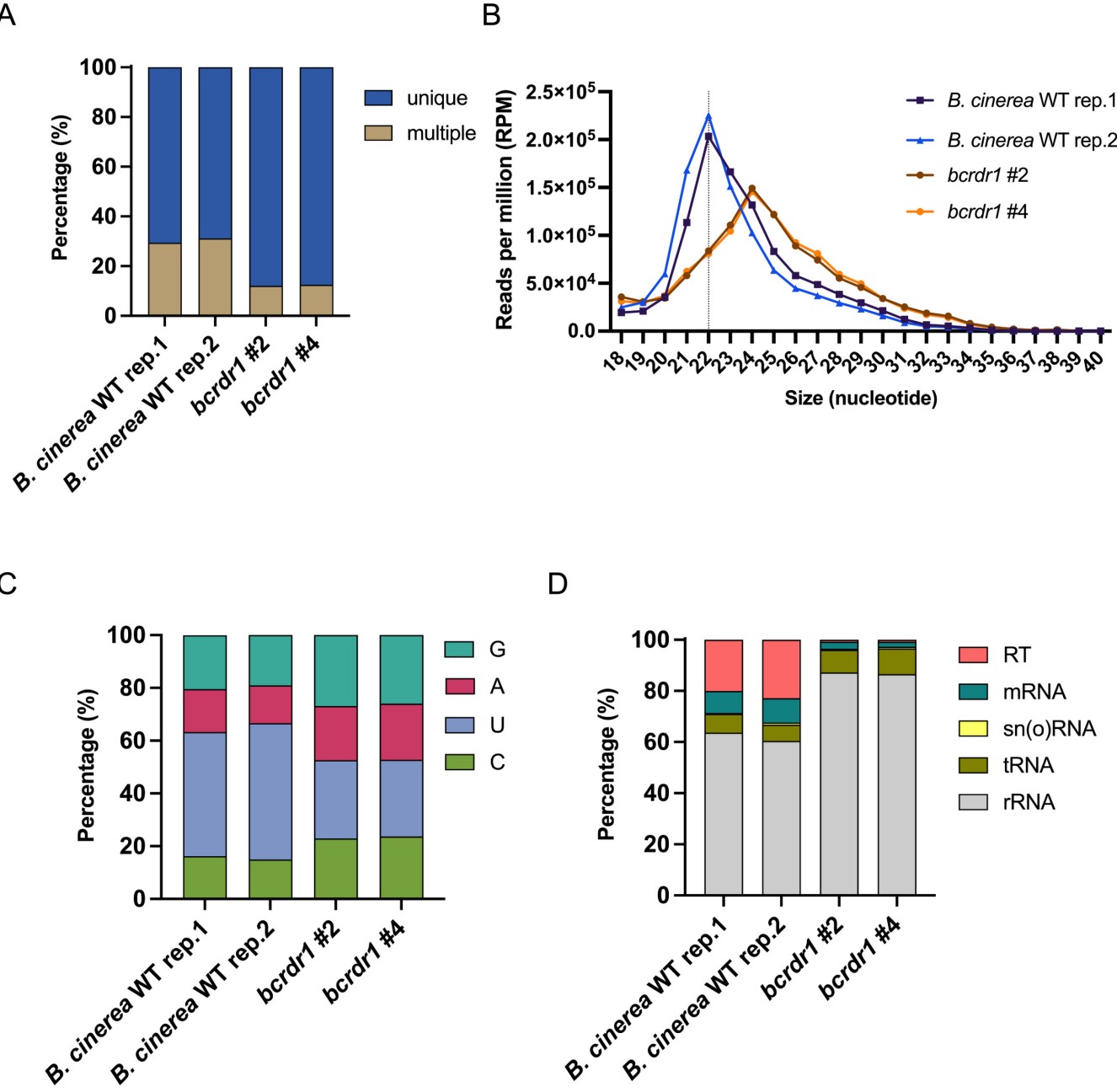

**Fig 2. Small RNA sequencing analysis of *B. cinerea* WT and *bcrdr1* ko mutants.** A) Fractions of Bc-sRNAs mapping unique or multiple times to a *B. cinerea* reference genome. B) Bc-sRNA size profiles (18–40 nt) of mapped Bc-sRNA reads in reads per million (RPM). C) Distribution in percentage of the four RNA nucleotides C, U, A, G at the 5'prime position of Bc-sRNAs. D) Distribution in percentage of Bc-sRNAs mapping to distinct *B. cinerea* RNA gene loci: ribosomal RNA (rRNA), transfer RNA (tRNA), small nuclear and nucleolar RNA (sn(o)RNA), messenger RNA (mRNA), and retrotransposon (RT).

oomycete pathogen of *A. thaliana*, *Hyaloperonospora arabidopsidis* [22]. In this adapted version, we fused the native target sites of Bc-siRNA3.1 and Bc-siR3.2 to the 5'prime or 3'ends of the *Csy4* transgene, respectively, turning *Csy4* into a target gene of these Bc-sRNAs. Csy4 constantly suppresses expression of the *GFP*, unless GFP expression is activated when Bc-sRNAs silence *Csy4*. This GFP switch-on reporter was stably expressed in transgenic *A. thaliana*

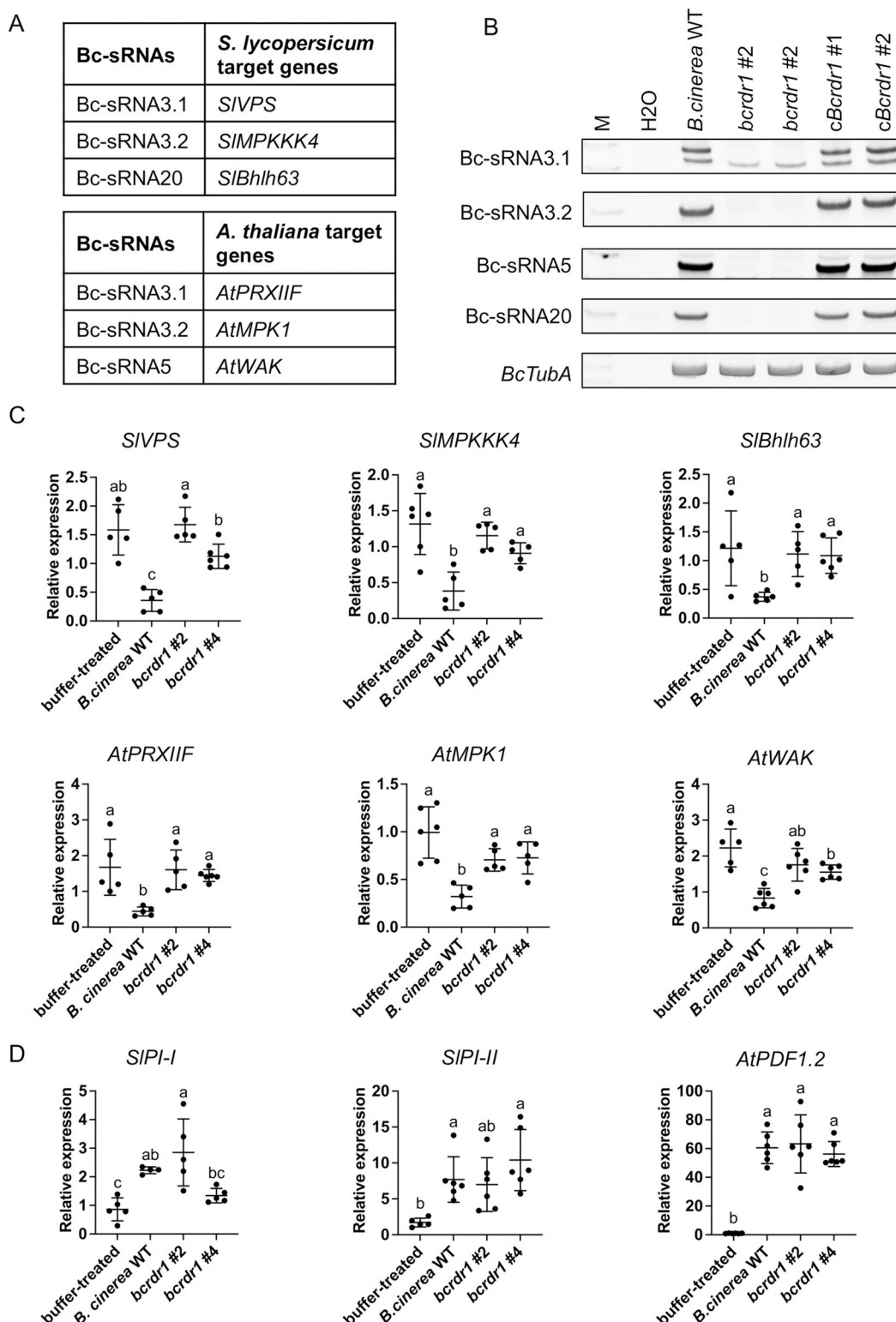

**Fig 3. *B. cinerea bcrdr1* ko mutants are compromised in plant target gene suppression.** A) Known *S. lycopersicum* and *A. thaliana* target genes silenced by Bc-sRNAs through cross-kingdom RNAi. B) Stem-loop reverse transcriptase PCR of Bc-sRNAs was carried out with tomato leaf samples infected with *B. cinerea* WT, *bcrdr1* ko mutants or cBcRDR1 strains at 48 hpi. Tomato leaves were inoculated with a 20 ml drop of $5 \times 10^4$/ml conidiospore suspension. *BcTubA* mRNA expression was used as an internal control. C) Quantitative reverse transcriptase PCR measuring mRNA levels of Bc-sRNA target genes

in *S. lycopersicum* and *A. thaliana* during infection with *B. cinerea* WT or *bcrdr1* ko mutants. D) Quantitative reverse transcriptase PCR measuring mRNA levels of immunity marker genes *SlPI-I*, *SlPI-II* and *AtPDF1.2*. In C) and D), *S. lycopersicum* leaves were inoculated with a 20 μl drop of 5 x $10^4$/ml conidiospore suspension and samples were collected at 36 hpi. *A. thaliana* leaves were inoculated with a 20 μl drop of 2 x $10^5$/ml conidiospore suspension and samples were collected at 60 hpi. Lines in scatter plots represent the mean and the standard deviation. Each gene was measured in at least four biological replicates. The *SlActin2* or *AtActin2* were used as reference genes. Statistical analysis was performed using ANOVA followed by a Tukey post-hoc test with *p*-value threshold $p < 0.05$.

plants. With these reporter plants, we were able to visualize *B. cinerea*-induced cross-kingdom RNAi in infected leaf tissue by fluorescent microscopy. Inoculating seedling leaves of reporter plants with *B. cinerea* WT conidiospore suspension led to enhanced GFP expression compared to leaves inoculated with the *bcrdr1* #2 ko mutant or treated with water (Figs 4B and S10). The non-invasive GFP switch-on reporter allowed us to quantify cross-kingdom RNAi by life-time imaging. We recorded a time lapse video of GFP activity in reporter plants either inoculated with *B. cinerea* WT or *bcrdr1* ko #2 for 45 hours (S1 Video). A clear increase in the GFP signal was apparent in *B. cinerea* WT infected leaves after 24 hpi (Figs 4C and S10). This signal was not related to plant auto-fluorescence, because no GFP signal was detected in *B. cinerea* WT-infected *A. thaliana* WT plants (S11 Fig). We verified the results obtained by fluorescence microscopy when measuring *GFP* mRNA and GFP protein levels by quantitative reverse transcriptase PCR and Western blot analysis using an anti-GFP antibody (Fig 4D and 4E). Using rosette leaves of adult *A. thaliana* plants, we confirmed among three independent infection series the activation of the GFP signal upon infection with *B. cinerea* WT, but not in leaves inoculated with the two *bcrdr1* ko #2 and #4 mutants or when infected with the previously characterized *bcdcl1/bcdcl2* double knock-out (*bcdcl1/2*) mutant strain (S12 Fig). The *bcdcl1/2* mutant is unable to produce reporter-activating Bc-sRNAs [13]. GFP activation was specific to *B. cinerea* infection, because infection with the oomycete pathogen of *A. thaliana*, *Hyaloperonospora arabidopsidis*, did not switch on the GFP reporter (Figs S13 and S14).

Referring to earlier in this study, we had observed that *bcrdr1* mutants developed less biomass during infection due to reduced virulence (Fig 1C). To rule out that reduced *B. cinerea bcrdr1* biomass could be responsible for lower GFP activity when inoculating the cross-kingdom RNAi reporter plants, we used 10x higher conidiospore suspension compared to *B. cinerea* WT inoculation and found that with increased conidiospore concentration, GFP activity was still significantly lower with *bcrdr1* ko (S15 Fig). Interestingly, GFP activity was the strongest at and beyond the infection front of *B. cinerea* (S15 and S16 Figs). This might indicate that cross-kingdom RNAi was the strongest in newly infected leaf cells and might spread into non-infected plant cell layers.

## *B. cinerea* small RNAs translocated into plants promote infection

Several independent studies revealed that knockout or knockdown of DCLs led to loss of Bc-sRNA biogenesis and reduced pathogenicity in *B. cinerea* and other fungal pathogens [11,13,23–29]. Here, we demonstrated that knocking out the *BcRDR1* leads to loss of Bc-sRNA production, compromises cross-kingdom RNAi, and reduces pathogenicity. However, *BcRDR1* deletion might have affected other endogenous small RNA regulatory processes in the fungus that are relevant for infection [30]. Therefore, we aimed to further validate that cross-kingdom RNAi was part of *B. cinerea* pathogenicity. Transgenes producing small RNA sponges, for example RNA short tandem target mimic (STTM), can block microRNA- and siRNA-induced silencing of plant endogenous and exogenous target genes [22, 31]. We cloned a Bc-siRNA3.2/Bc-sRNA5 double STTM (Fig 5A) and transformed it into *A. thaliana* for stable expression. We could isolate three independent *A. thaliana* STTM T2 lines and infected

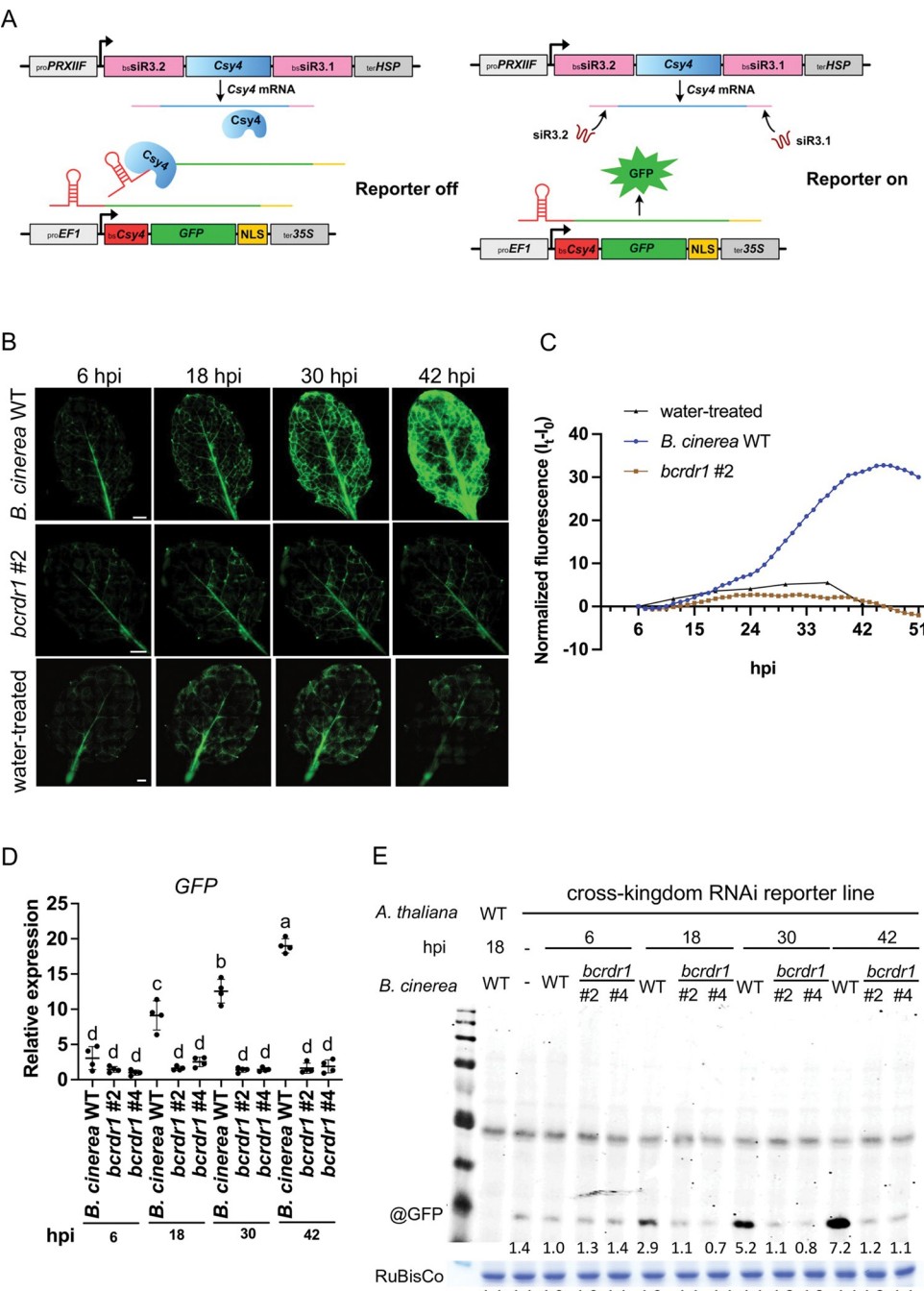

**Fig 4. *B. cinerea bcrdr1* ko mutants are compromised in cross-kingdom RNAi.** A) Schematic overview of a GFP-based switch-on cross-kingdom RNAi reporter suitable for *in planta* expression. B) Fluorescence microscopy images from *B. cinerea* WT, *bcrdr1* #2 or water-treated GFP reporter plant seedlings at different time points. A 5 µl drop of a 5 x 10⁵/ml conidiospore suspension was placed at the center of the leaf before placing a glass covering slip on the top that dispersed the conidiospore suspension over the entire leaf surface. The scale bars represent 1 mm. C) GFP quantification in WT and *bcrdr1* ko #2-infected seedling leaves or water-treated leaves from 6–51 hpi. For normalization, GFP fluorescence signal intensity at different time points ($I_t$) was subtracted with the initial GFP signal intensity ($I_0$). D) Quantitative reverse transcriptase PCR analysis of relative *GFP* mRNA expression in cross-kingdom RNAi reporter plants during *B. cinerea* infection. *AtActin2* was used as a reference gene. Statistical analysis was performed using ANOVA followed by a Tukey post-hoc test with *p*-value threshold $p < 0.05$. The "-"symbol represents water-treated leaves. E) Western blot analysis of GFP expression in cross-kingdom RNAi reporter plants during *B. cinerea* infection using a @GFP antibody. The ribulose 1,5-bisphosphate carboxylase/oxygenase (RuBisCo) signal detected by Coomassie Brilliant Blue (CBB) staining was used as a loading control. Numbers indicate GFP and RuBisCo signal intensities estimated by the FIJI software.

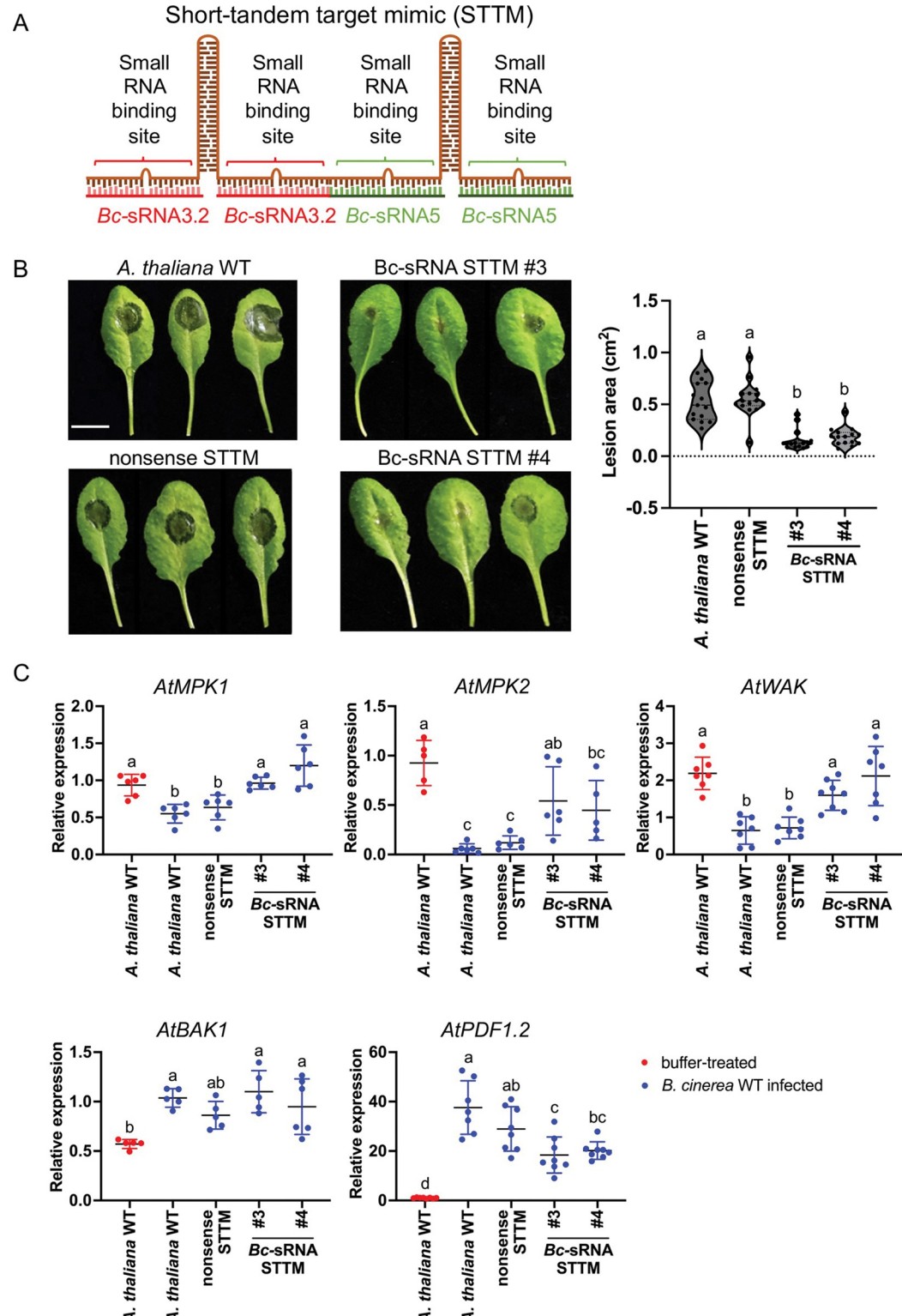

**Fig 5. A plant expressing Bc-sRNA STTM blocks cross-kingdom RNAi and reduces *B. cinerea* pathogenicity.** A) Schematic overview of the STTM construct to block Bc-sRNA3.2 and Bc-sRNA5 *in planta*. B) *A. thaliana* detached leaf inoculation assay using a 20 µl drop of 2 x 10$^5$/ml conidiospore suspension of *B. cinerea* WT, comparing three independent Bc-sRNA STTM T2 plant lines (#3, #4, #6) with *A. thaliana* WT and a nonsense STTM T2 plant line serving as negative controls. Lesion sizes were measured at 60 hpi and a minimum 15 lesions per plant line were used for statistical analysis

using ANOVA followed by a Tukey post-hoc test with *p*-value threshold $p < 0.05$. The scale bars represent 1 cm. C) Quantitative reverse transcriptase PCR measuring mRNA levels of Bc-sRNA target genes in *A. thaliana AtMPK1*, *AtMPK2*, and *AtWAK* comparing STTM lines and WT during *B. cinerea* infection of WT plants (red dots) or STTM-expressing plants (blue dots) at 48 hpi. Lines in scatter plots represent the mean and the standard deviation. Each gene was measured in at least five biological replicates. The *At-Actin* was used as a reference gene. *AtBAK1* and *AtPDF1.2* were measured as plant immunity marker genes induced during *B. cinerea* infection. Statistical analysis was performed using ANOVA followed by a Tukey post-hoc test with *p*-value threshold $p < 0.05$.

those lines with *B. cinerea* WT. Bc-sRNAs STTM plants exhibited reduced lesion sizes induced by *B. cinerea*, compared to *A. thaliana* WT or a transgenic *A. thaliana* line expressing a non-sense RNA-STTM (Figs 5B and S17A). In consistence, Bc-sRNA3.2 and Bc-sRNA5 target genes in *A. thaliana*, *AtMPK1*, *AtMPK2*, and *AtWAK* were suppressed in *A. thaliana* WT or *A. thaliana* expressing a nonsense RNA STTM upon *B. cinerea* infection, but suppression of those host target genes was abolished in the Bc-sRNA STTM plant lines (Figs 5C and S17B). Successful infection of all plant lines resulted in an induced host defense response due to *B. cinerea* infection, which was validated by measured up-regulation of the *A. thaliana* immunity-related genes *AtBAK1* and *AtPDF1.2*. These results confirmed that cross-kingdom RNAi is an important part of *B. cinerea* pathogenicity.

We previously found that expressing a triple STTM in *A. thaliana* that blocked small RNAs of the oomycete pathogen *Hyaloperonospora arabidopsidis* led to reduced disease levels [22]. We hypothesized that combining binding sites of different types of pathogen and parasite small RNAs that are known to induce cross-kingdom RNAi might create a multi-pathogen resistant plant genotype. We designed a "master" STTM to block the two *B. cinerea* Bc-sRNA3.2 and Bc-siRNA5, the two *H. arabidopsidis* small RNAs, Hpa-sRNA2 and Hpa-sRNA90 [22], as well as the two *Cuscuta campestris* ccm-miR12497b and ccm-miR12480 [32] and the *Phytophthora infestans* Pi-miRNA8788 [33]. *A. thaliana* master STTM plants grew and developed normally and did not show any pleiotropic defect (S18 Fig). We inoculated three independent T2 master STTM plant lines with *B. cinerea* and found that all transgenic lines exhibited reduced fungal lesion sizes (S17A and S17B Fig) and plant target gene de-repression (S17C Fig), confirming our results obtained with the Bc-sRNA STTM plants. Interestingly, a master STTM plant also revealed reduced disease levels when inoculated with the oomycete *H. arabidopsidis* (S17E Fig). Reduced infection was not due to constantly enhanced plant immunity, according to non-induced expression of *AtPR1* or *AtPFD1.2* in non-treated plants (S17D Fig). Further, inoculating Bc-sRNA- or master STTM plant lines with the bacterial pathogen *Pseudomonas syringae* DC3000 did not result in reduced bacterial colony numbers (S17F Fig). Based on these observations, we propose that STTMs might be a future application against pathogen small RNAs to design multi-pathogen resistant plants.

## Discussion

In this study, we demonstrated that BcRDR1 is a pathogenicity factor that is required for cross-kingdom RNAi in the fungal plant pathogen *B. cinerea*. Previously, a *bcdcl1/2* mutant was characterized to be impaired in pathogenicity, Bc-sRNA biogenesis, and cross-kingdom RNAi [13,26]. This *dcl1dcl2* ko mutant exhibited reduced growth in axenic culture as well as on plant leaves that could be partially complemented by ectopic Bc-sRNA expression in transgenic *A. thaliana* [13]. Effective *BcDCL1/BcDCL2* gene knockdown via both host-induced gene silencing or RNA spray-induced gene silencing was sufficient to reduce *B. cinerea* disease symptoms on various plant species and tissues supporting a functional role of BcDCLs in pathogenicity [23,24,26]. Recently, the impact of *bcdcl1dcl2* ko generated in the *ku70* ko background was debated to not exhibit any growth defect in axenic culture and no or only mildly

reduced pathogenicity on plant leaves [34,35]. The *bcrdr1* ko mutants characterized here did not display any growth defect when grown in axenic culture, but a significant reduction in lesion size formation during plant infection and reduced *in planta* fungal biomass when infecting the two plant host species *S. lycopersicum* and *A. thaliana*. Furthermore, *bcrdr1* ko mutants were impaired in activating a Bc-sRNAs-responsive switch-on cross-kingdom RNAi reporter expressed in transgenic *A. thaliana*. Moreover, we confirmed with further evidence the importance of cross-kingdom RNAi in *B. cinerea* pathogenicity using a STTM approach. Blocking Bc-sRNAs via STTM expression in *A. thaliana* led to reduced *B. cinerea* disease symptoms. A similar approach demonstrated the relevance of cross-kingdom RNAi in the oomycete pathogen *H. arabidopsidis* infecting *A. thaliana* [22].

*RDR* knock out in different fungal plant pathogens resulted in pleiotropic defects on growth, development, and pathogenicity. The rice blast fungus *M. oryzae* comprises three RDRs named MoRDRP1-MoRDRP3, and the MoRDRP1 is the closest orthologue of BcRDR1. Isolated *mordrp1* ko mutants exhibited reduced growth, conidia formation, and disease symptoms on rice leaves [10]. Unlike *bcrdr1*, *mordrp1* revealed only moderate changes in small RNA production, but *mordrp2* ko was strongly impaired in retrotransposon- and intergenic-derived 21–23 nt small RNA production. Thus, the underlying MoRDRP1 mode of action in pathogenicity remained obscure. In the head blight-inducing fungal pathogen *F. graminearum* five RDRs, named FgRdRP1-FgRdRP5, were identified. In two independent studies, distinct roles of FgRdRPs in fungal development and pathogenicity were reported [11,36]. Studying *fgrdrp* ko mutants revealed neither altered scab disease symptoms when infecting flowering wheat heads or rot symptoms on tomato fruits, nor defects in vegetative development under normal or abiotic stress growth conditions [36]. On the contrary, different *fgrdrp* ko mutants displayed alterations in asexual and sexual development. Moreover, infecting wheat spikes with *fgrdrp2*, *fgrdrp3* and *fgrdrp4* ko mutants at 9 days post inoculation resulted in reduced head blight symptoms, which correlated with lower pathogen DNA content and levels of the mycotoxin Deoxynivalenol per seed dry weight [11]. The role of FgRdRPs in small RNA biogenesis has not been studied so far. The alternating numbers of RDR homologs in different fungal species and their distinct roles in small RNA biogenesis and pathogenicity indicate a rapid RDR neo-functionalization of the RNAi components in fungi. The unknown roles of BcRDR2 and BcRDR3 in *B. cinerea* need to be investigated in future studies.

Movement of small RNAs between fungi and plants has been observed multiple times [37]. Cross-kingdom RNAi has been discovered in other plant- or animal-associated fungal, oomycete, and bacterial microbes, for pathogens and symbionts [14,16]. It will be interesting to investigate whether microbial RDRs play a broader role in diverse plant-microbe interactions and in cross-kingdom RNA communication and its practical implementation by RNAi-based crop protection strategies.

## Materials and methods

### Fungal and plant materials and growth conditions

*Botrytis cinerea* (Pers.: Fr.) strain B05.10 was used for this study. Standard cultivation was carried out on HA medium (10 g/L malt extract, 4 g/L yeast extract, 4 g/L glucose, 15 g/L agar). *B. cinerea bcrdr1* ko and *bcdcl1dcl2* ko mutant strains were grown on HA medium supplemented with 70 µg/mL hygromycin B (Carl Roth GmbH). Fungal plates were incubated at room temperature in a growth chamber under long wavelength UV light (EUROLITE; 20 W to stimulate sporulation. The tomato (*Solanum lycopersicum* (L.) cultivar Heinz) used in this study was grown under the condition of 24°C, 16 h light/8 h dark, 60% humidity in a growth cabinet. *Arabidopsis thaliana* (L.) ecotype Col-0 was grown under short day condition (10 h light/ 14 h

dark, 22˚C, 60% relative humidity) for *Botrytis* infection. For *H. arabidopsidis* infection, *Arabidopsis* Col-0 was grown under long day condition (16 h light/ 8 h dark, 22˚C, 60% relative humidity). Transgenic T1 and T2 *A. thaliana* lines were selected on 1/2 Murashige and Skoog (MS) medium supplemented with kanamycin (Carl Roth; 100 µg/mL). Independent plant transformant lines were indicated (e.g., STTM #3).

### Cloning of *bcrdr* ko and cBcRDR1 complementation vectors, plant STTMs and cross-kingdom RNAi reporter

The Golden Gate cloning strategy was used for cloning fungal gene ko and gene complementation cassettes as well as plant expression vectors following the instruction, as described in Binder *et al.* [38]. The STTM sequences were designed, as described previously [31], and flanking region with BsaI recognition sites were introduced. A previously designed transgenic *A. thaliana* line that expressed a STTM with a randomized sequence of *H. arabidopsidis* small RNA target site [22] was used as a nonsense STTM control in this study.

For *A. thaliana* reporter lines, a Csy4 coding sequence was synthesized by MWG Eurofins with codon optimization for expression in plants, and the reporter cassette were assembled, as previously described [22]. The native promoter of the Bc-sRNA3.1 target gene *AtPRXIIF* was used to control the transcription level of the *Csy4* transgene in order to mimic natural Bc-sRNA target expression.

### Fungal and plant transformation

*B. cinerea* was transformed as previously described [39] with minor modifications. Transformed fungal protoplasts were mixed with SH agar (0.6 M sucrose, 5 mM Tris-HCl (pH 6.5), 1 mM $(NH_4)H_2PO_4$, 8 g/L agar) without any antibiotics and incubated in darkness for 24 hours. A second layer of fresh SH agar containing hygromycin B (25 µg/ml) was added to the top after pre-incubation. The plates were further incubated in darkness until isolation of fungal transformants. For *A. thaliana* transformation, a previously described floral dip method was used [40].

### Plant infection assay

*S. lycopersicum* pathogenicity assays were performed on detached leaves from 4- to 5-week-old plants. *B. cinerea* conidia were resuspended in 1% malt extract at a final concentration of $5×10^4$ conidia/ml. *S. lycopersicum* leaves were inoculated with 20 µl conidia suspension and inoculated in a humidity plastic box. The detached leaves were placed on moist filter paper and then incubated in a closed plastic box. *A. thaliana* pathogenicity assays were performed on detached leaves from 5-week-old plants by inoculation with $2 × 10^5$ conidia/ml *B. cinerea* conidia resuspended in 1% malt extract. 15 µl conidia suspension was dropped at the center of each leaf and inoculated in a humidity plastic box. Infected leaves were photographed, and lesion area was measured using the Fiji software (ImageJ version 2.1.0/1.53c).

*Hyaloperonospora arabidopsidis* (GÄUM.) isolate Noco2 was maintained on *A. thaliana* Col-0 WT plants. Two-week-old plants were inoculated with $2×10^4$ conidia/ml suspension. Samples were harvested at 7 dpi into 10 ml of sterile water. The sporangiophore numbers were counted on detached cotyledons using a binocular microscope.

*Pseudomonas syringae* pv. *tomato* DC3000 was streaked out on LB agar plates with Rifampicin for 2 days at 28˚C. A single colony from the plate was inoculated with LB liquid medium with Rifampicin overnight. *Pseudomonas* cells were harvested and re-suspended in 10 mM $MgCl_2$ with 0.04% Silwet L-77 and adjusted to $OD_{600}$ = 0.02. 5-week-old *A. thaliana* plants were sprayed with *Pseudomonas* suspension. Samples were harvested at 3 dpi and 4 leaf discs

per plant were collected then homogenized in 10 mM MgCl$_2$ for one biological replicate. A serial dilution on LB agar plates with Rifampicin was performed to count colony forming units.

## Microscopic analysis of the switch-on GFP cross-kingdom RNAi reporter

For recording GFP time infection course with switch-on in cross-kingdom RNAi reporter plants, 3-week-old *A. thaliana* seedlings were cultivated on ½ MS + 1% sucrose agar plates and inoculated on plates with 2x10$^5$ conidiospore/ml *B. cinerea* resuspended in 1% malt extract medium pre-incubated for 1 hours. The pre-incubated conidiospore suspension was washed twice with sterile water before 5 µl was dropped at the center of one leaf per seedling. For time course video, the Leica DMi8 Thunder Imager equipped with a Leica DFC9000 GT camera was used and whole seedlings were imaged. Imaging was set to start 6 h post conidiospore inoculation and images were taken in cycles of 22.5 mins for 45 h in total. Raw imaging data were processed using the Leica LAS X microscope software. Video editing was performed using Adobe Premiere Pro CC version13.0. GFP fluorescence intensity raw data were normalized at given time points ($I_t$) by subtracting the initial GFP intensity ($I_0$).

For the documentation of GFP signals in adult reporter plants, detached rosette leaves from 5-week-old *A. thaliana* plants were inoculated with $2 \times 10^5$ conidia/ml *B. cinerea* conidia resuspended in 1% malt extract for 1 hour. Conidiospore suspension was washed with sterile water and 20 µl was dropped on the leaf. Single leaf images of GFP signals and bright field (BF) were recorded on a Leica DM6 B upright microscope equipped with a Leica DFC9000 GT camera.

## Trypan Blue staining

Leaf samples were collected and stained with Trypan Blue solution as described previously [41]. Microscopic images were taken with a DFC450 CCD-Camera (Leica) on a CTR 6000 microscope.

## GUS staining

Leaf samples were vacuum-infiltrated with GUS staining solution (0.5 mg/ml X-Gluc, 100 mM phosphate buffer pH 7.0, 10 mM EDTA pH 7.0, 1 mM K3[Fe(CN)6], 1 mM K4[Fe(CN)6], 0.1% Triton X-100) and incubated over night at 37˚C. Leaves were de-stained with 70% ethanol overnight and microscopic images were taken under DFC450 CCD-Camera (Leica) on a CTR 6000 microscope.

## Genotyping PCR

A CTAB method followed by chloroform extraction and isopropanol precipitation was used for DNA extraction from fungal mycelium [42]. The GoTaq G2 Polymerase (Promega) was used for genotyping PCR using PCR primers as listed in S1 Table.

## Quantitative PCR

For fungal biomass quantification, genomic DNA was isolated using the CTAB method [43]. Relative fungal biomass was estimated by qPCR using the *BcTubA* primers (S1 Table). Raw data were normalized to plant genomic DNA estimation using the *AtActin2* or *SlActin2* primers (S1 Table).

For gene expression analysis, a CTAB method was used for total RNA extraction [44]. Genomic DNA was removed by DNase I (Sigma-Aldrich) treatment following the

manufacturer's instruction. 1 μg of total RNA from each sample was used for cDNA synthesis using the SuperScriptIII reverse transcriptase (ThermoFisher Scientific). Genomic DNA quantities and gene expression were measured by quantitative real-time PCR using the Primaquant low ROX qPCR master mix (Steinbrenner, Laborsysteme). Differential gene expression level was calculated using the $2^{-\Delta\Delta Ct}$ method [45]. Primers used in quantitative RT-PCR are listed in S1 Table and raw data are given in S3 Table.

## Stem-loop reverse transcription PCR

Small RNA detection by stem-loop reverse transcriptase PCR was carried out following the protocol, as described [46]. 1 μg of total RNA was used for small RNA-specific RT and PCR. Reverse transcriptase PCR products were separated on a 10% non-denaturing polyacrylamide gel followed by ethidium bromide staining. Primers used in stem-loop reverse transcriptase PCR are listed in S1 Table.

## Small RNA sequencing

Small RNAs were isolated from total RNA extracts using 15% polyacrylamide gel electrophoresis. Isolated small RNAs were subjected for library cloning using the Next Small RNA Prep kit (NEB) and sequenced on an Illumina NextSeq 2000 platform. The Illumina sequencing data were analyzed using the GALAXY Biostar server [47]. Raw data were de-multiplexed (Illumina Demultiplex, Galaxy Version 1.0.0) and adapter sequences were removed (Clip adaptor sequence, Galaxy Version 1.0.0). Sequence raw data are deposited at the NCBI SRA server (BioProject ID PRJNA978613). Reads were then mapped to the reference genome assembly of *B. cinerea* (ASM14353v4) using the BOWTIE2 algorithm (Galaxy Version 2.3.4.2) in end-to-end alignment mode, without setting–k or–a options. To assign Bc-sRNA to different types of RNA encoding genes, the sequence information of *B. cinerea* ribosomal RNAs, transfer RNAs, small nuclear and nucleolar RNAs, and mRNA were downloaded from the Ensembl database. Retrotransposon sequences was used as annotated in a previous study [17]. Reads were counted and normalized on total *B. cinerea* reads per million (RPM).

Small RNA coverage plots were produced from the above sRNA alignment, separating coverages by alignment strand and read length (grouping reads < 20 and > 25 nucleotides long). Coverage values represent the cumulative depth of the separate groups at every read position, normalized by aligned reads per million (RPM).

## RDR homology search

*Neurospora crassa* QDE-1 and SAD-1 amino acid sequences (S2 Table) were used for Blastp search against the *B. cinerea* strain B05.10 (Taxon ID: 332648) protein reference database in Uniprot. RDR Active sites alignment was conducted with CLC Main Workbench (version 20.0.4).

## Phylogeny

CLC Main Workbench (version 20.0.4) was used for phylogenetic analysis and composing the DNA sequence labels. Alignment was conducted with amino acid sequences of each RDR by default multiple alignment algorithms. Gap open cost was set for 15.0 and gap extension cost was set as 1.0. Phylogenetic tree was carried out by neighbor joining algorithm with 2000 replicates bootstrap. Kimura protein distance is used for protein distance measurement.

## Data plotting and statistical analysis

GraphPad Prism 9 software was used for plotting and statistical analysis. For multi-samples comparison, one-way ANOVA with Tukey multiple comparisons test (p-value < 0.05) was carried out for data analysis. For *H. arabidopsidis* infection assay, unpaired t-test was carried out. Statistical significance was set for two-tailed p-value < 0.05 (*), p-value < 0.01 (**).

## Data accessibility

Sequencing data have been deposited in NCBI SRA (BioProject ID PRJNA978613).

## Supporting information

**S1 Fig.** Fungal RDR phylogenetic analysis (A) and amino acid sequence alignment of the RDR active site (B). The bar in (A) represents length of branch. Min and max refers to levels of conservation in (B). Amino acids sequences used in this analysis are given in S2 Table.
(PDF)

**S2 Fig. Expression levels of the three *BcRDRs*.** *BcRDR*s mRNA levels were measured in two independent replicates by qRT-PCR using the *BcTubA* as a reference gene. Lines in scatter plots represent the mean and the standard deviation. Statistical analysis was performed using ANOVA followed by a Tukey post-hoc test with *p*-value threshold $p < 0.05$.
(PDF)

**S3 Fig. Generation of *bcrdr* ko mutants and cBcRDR1 gene complementation strains.** A) Schematic overview of the *bcrdr1* ko and *cBcRDR1* complementation cloning strategies. B) Genotyping PCRs assessing *bcrdr1* gene ko and the insertion of the ko cassette into the *BcRDR1* genomic context. C) Genotyping PCRs assessing the insertion of the *cBcRDR1* cassette into the *bcrdr1* ko genomic context and RT-PCR assessing expression of *BcRDR1* in WT, *bcrdr1* ko mutant and cBcRDR1 complementation strains. D) Schematic overview of the *bcrdr2* ko cloning strategy. E) Genotyping PCR assessing *bcrdr2* gene ko.
(PDF)

**S4 Fig.** Replicates of infection series with *B. cinerea* WT and *bcrdr1* ko mutants on detached *S. lycopersicum* (A) and *A. thaliana* (B) leaves. A 20 μl drop of 5 x 10⁴/ml condidiospores was placed on the leaves. The scale bars represent 1 cm. Lesion size induced by *B. cinerea* infection was measured at 48 hpi. Numbers given the analyzed lesions per plot. Statistical analysis was performed using ANOVA followed by a Tukey post-hoc test with *p*-value threshold $p < 0.05$.
(PDF)

**S5 Fig. *B. cinerea* WT, *bcrdr1* ko, *bcrdr2* ko mutants and cBcRDR1 growth on 1% malt extract agar.** A) Plate growth images were taken at 4 days. The scale bars represent 1 cm. B) To measure growth curves, a drop of 2 x 10⁵/ml 10 μl conidiospore suspension was placed at the center of the agar plate and colony diameter was measured at 3, 4 and 5 days. Data represent 5 replicates. Statistical analysis was performed using ANOVA followed by a Tukey post-hoc test with *p*-value threshold $p < 0.05$.
(PDF)

**S6 Fig. Expression analysis of known *B. cinerea* virulence genes in the *bcrdr1* ko mutants.** mRNA levels of *BcPG1* (Bcin14g00850), *BcNEP1* (Bcin06g06720), *BcSpl1* (Bcin03g00500), *BcXyn11A* (Bcin03g00480) and *BcHIP1* (Bcin14g01200) was compared in *B. cinerea* WT and the *bcrdr1* ko mutants #2 and #4 grown in axenic culture in four biological replicates. The *BcTubA* was used as a reference gene. Statistical analysis was performed using ANOVA

followed by a Tukey post-hoc test with *p*-value threshold $p < 0.05$.
(PDF)

**S7 Fig.** A) Mapping results at the *BcGyspy1* and *BcGypsy3* loci obtained from *B. cinerea* WT and *bcrdr1* ko mutants small RNA sequencing data. Reads per million (RPM) values > 0 indicate sense alignment, RPM values < 0 indicates antisense alignment. Color-code indicates Bc-sRNA sizes. B) Size profiles of Bc-sRNAs mapped to the *BcGypsy1* and *BcGypsy3* loci in *B. cinerea* WT or *bcrdr*1 ko mutants #2 and #4. C) Expression levels of *BcGypsy1* and *BcGypsy3* mRNAs in *B. cinerea* WT and *bcrdr1* ko mutants when grown in axenic culture. D) Expression levels of *BcGypsy1* and *BcGypsy3* mRNAs in *B. cinerea* WT grown under axenic culture condition and during tomato infection at 48 hpi. E) Expression level comparison of *BcGypsy1* and *BcGypsy3* mRNAs in *B. cinerea* WT versus *bcrdr1* ko mutants #2 and #4 when grown under axenic culture condition or during tomato infection at 48 hpi. In C), D) and E), *BcTubA* mRNA was used as a reference gene expression. Lines in scatter plots represent the mean and the standard deviation. Statistical analysis was performed using ANOVA followed by a Tukey post-hoc test with *p*-value threshold $p < 0.05$.
(PDF)

**S8 Fig. Stem-loop RT-PCR of Bc-sRNAs comparing *B. cinerea* WT and *bcrdr1* ko mutants.** Bc-sRNAs were detected in tomato leaf samples infected with *B. cinerea* at 48 hpi. Figure represents full-scale gel images of results, as given in Fig 3B.
(PDF)

**S9 Fig.** Biological replicates of mRNA expression measurements of known Bc-sRNA target genes in *S. lycopersicum* (A) and *A. thaliana* (B) during infection with *B. cinerea* WT and *bcrdr1* ko mutants. Samples were taken at 36 hpi or 60 hpi for *S. lycopersicum* and *A. thaliana*, respectively. The *SlActin2* or the *AtActin2* were used as reference genes. Lines in scatter plots represent the mean and the standard deviation. Statistical analysis was performed using ANOVA followed by a Tukey post-hoc test with *p*-value threshold $p < 0.05$.
(PDF)

**S10 Fig. Independent infection time series using GFP switch-on cross-kingdom RNAi reporter plant seedlings infected with *B. cinerea* WT, *bcrdr1* ko mutant #2 or water-treated.** A) Fluorescence microscopy images at different time points of infection. A 5 μl drop of a $2 \times 10^5$/ml conidiospore suspension was placed at the center of the leaf before placing a glass covering slip on the top that dispersed the spore suspension and led to GFP activation over the entire leaf. The scale bars represent 1 mm. B) Normalized GFP signal quantification of whole seedling leaves over the time series of 6–42 hpi.
(PDF)

**S11 Fig. Fluorescence microscopic imaging of *A. thaliana* WT seedlings infected with *B. cinerea* WT.** The scale bars represent 1 mm.
(PDF)

**S12 Fig. Three independent infection series and fluorescence microscopic imaging using detached rosette leaves of adult GFP switch-on cross-kingdom RNAi reporter plants.** Leaves were inoculated with a 20 μl drop of $2 \times 10^5$/ml conidiospore suspension of *B. cinerea* WT, *bcrdr1* ko mutants, and a *bcdcl1/2* mutant. *A. thaliana* WT plants were infected with *B. cinerea* WT to assess auto-fluorescence, and water-treated GFP reporter plants were assessed for reporter auto-activity. The scale bars represent 500 μm.
(PDF)

**S13 Fig. Infection of the GFP reporter seedling plants with the oomycete pathogen *Hyaloperonospora arabidopsidis*.** 10 ml of a 2 x $10^4$/ml conidiospore suspension was sprayed onto leaves. Trypan Blue staining visualized oomycete hyphae in the infected leaf. Outlines indicate the same leaf area in fluorescence and Trypan Blue staining images.
(PDF)

**S14 Fig. Infection of a transgenic *A. thaliana* cross-kingdom RNAi reporter line with a GUS reporter.** This reporter line was previously designed to demonstrate cross-kingdom RNAi triggered by small RNAs secreted by the oomycete *H. arabidopsidis* (Dunker *et al.*, 2020 [22]). A) Infection of the GUS reporter line with *H. arabidopsidis* showing GUS at infecting oomycete hyphae. B) Infection of the GUS reporter line with *B. cinerea* revealed no GUS activation at infection sites (indicated by a turquoise arrows). C) Infection of a GUS reporter line carrying scrambled small RNA target sites with *H. arabidopsidis* showing no GUS activity. D) Infection of the scrambled GUS reporter line with *B. cinerea* showing no GUS activity at infection sites (indicated by a turquoise arrows). For *B. cinerea* inoculation, a 20 μl drop of 2 x $10^5$ conidiospores were placed onto leaves. For *H. arabidopsidi*s infection, 10 ml of a 2 x $10^4$/ml conidiospore suspension were sprayed onto leaves. The scale bars represent 100 μm.
(PDF)

**S15 Fig. Fluorescence microscopy imaging, mRNA and protein expression measurement with adult rosette leaves of the GFP switch-on cross-kingdom RNAi reporter plant using 10x conidiospore concentration for *bcrdr1* inoculation.** A) Fluorescence microscopy images at 24 hpi indicated enhanced GFP expression in reporter plants at infection sites of *B. cinerea* WT in contrast to GFP non-expressing *A. thaliana* WT plants, water-treated GFP reporter plants or GFP reporter plants infected with *bcrdr1* ko mutants. The turquoise arrow in B) indicates the infection front of the *B. cinerea* WT inoculation. The scale bars in A) and B) represent 500 μm. C) GFP mRNA expression levels in *A. thaliana* WT plants (red dots), *A. thaliana* GFP reporter plants (blue dots) infected with *B. cinerea* WT, *bcrdr1* ko mutants or water-treated. The *AtActin2* was used as a reference gene. D) Western blot analysis of GFP expression using a @GFP antibody. RuBisCo signals were visualized by Coomassie Brilliant Blue staining. Numbers indicate GFP and RuBisCo intensities estimated by the FIJI software. E) *B. cinerea* genomic DNA in infected *A. thaliana* WT or GFP reporter plants was measured by qPCR using primers of the *BcTubA* gene. Raw data were normalized to plant DNA using *AtActin2* primers. Lines in scatter plots of qRT-PCR data in C) and qPCR of *B. cinerea* genomic DNA in E) represent the mean and the standard deviation. Statistical analysis was performed using ANOVA followed by a Tukey post-hoc test with *p*-value threshold $p < 0.05$.
(PDF)

**S16 Fig. Fluorescence microscopy imaging of adult rosette leaves from the GFP switch-on cross-kingdom RNAi reporter plants infected with *B. cinerea* WT at 36 hpi.** *B. cinerea* mycelium was visualized by Trypan Blue staining. Squares in images indicate area of magnification. The red asterisks indicate the same leaf trichome in merged BF/GFP and Trypan Blue images. The turquoise arrow indicates *B. cinerea* mycelium. The scale bars represent 500 μm.
(PDF)

**S17 Fig. Infection series with *B. cinerea*, *H. arabidopsidis* and *P. syringae* DC3000 in transgenic T2 *A. thaliana* lines expressing a Bc-sRNA or master STTM.** A) Leaf images of *A. thaliana* STTMs upon *B. cinerea* infection with 2 x $10^5$ conidiospores/ml at 60 hpi. The scale bar in represent 1 cm. B) Lesion area induced by *B. cinerea* infection was measured at 48 hpi. C) mRNA expression levels of Bc-sRNA target genes *AtMPK1* and *AtWAK* in *A. thaliana*. The *AtActin2* was used as a reference gene. Lines in scatter plots represent the mean and the

standard deviation. Statistical analysis was performed using ANOVA followed by a Tukey post-hoc test with $p$-value threshold $p < 0.05$. D) Semi-quantitative RT-PCR of the *A. thaliana* immunity-associated genes *AtPR1* and *AtPDF1.2*. *B. cinerea*-infected leaves were used as an immunogenic control. E) Infection of the master STTM line #3 with the oomycete *H. arabidopsidis*. Oomycete sporangiophores were counted at 7 dpi in three replicated inoculation experiments. F) Infection of *A. thaliana* STTM lines with the bacterial pathogen *Pseudomonas syringae* DC3000. Colony-forming units (cfu) of were counted at 3 dpi. Statistical analysis in B), C), E), F) was performed using ANOVA followed by a Tukey post-hoc test with $p$-value threshold $p < 0.05$. Statistical analysis in E) replicate #1 and replicate #2 was carried out by unpaired t-test with two-tailed p-value $< 0.05$ (*), p-value $< 0.01$ (**).
(PDF)

**S18 Fig. Plant images showing unaltered growth phenotype of *A. thaliana* STTM plants.** Pictures were taken at 38 days after growing in short-day condition. The scale bar represents 1 cm.
(PDF)

**S1 Video.** Time course of the GFP cross-kingdom RNAi reporter activity upon *A. thaliana* leaf inoculation from 6–51 hpi with *B. cinerea* WT (left site) or *bcrdr1* ko mutant #2 (right site).
(MP4)

**S1 Table. DNA oligonucleotides used in this study.**
(XLSX)

**S2 Table. Amino acid sequences used in phylogeny analysis.**
(XLSX)

**S3 Table. qPCR and qRT-PCR raw data $2^{-\Delta\Delta Ct}$ values.**
(XLSX)

## Acknowledgments

We thank Dr. Claude Becker for critical proofreading of this work. We want to thank the Gene Center Munich for Illumina NextSeq sequencing service. We would like to thank Dr. Martin Parniske for scientific discussions and providing access to the Golden Gate cloning system, Dr. Silke Robatzek and Dr. Eliana Mor for the access to and technical assistance with the DMi8 Thunder Imager microscope, and Dr. Dagmar Hann for sharing with us the *Pst* DC3000 strain. We thank Annika Lübbe for supporting the GUS staining. We thank Verena Klingl for technical support to isolate *Botrytis bcrdr1* ko transformants and Ignacio Mohr for helping with the *H. arabidopsidis* inoculation. We thank Franz Oberkofler for video editing.

## Author Contributions

**Conceptualization:** Arne Weiberg.

**Data curation:** An-Po Cheng, Nathan R. Johnson.

**Formal analysis:** An-Po Cheng, Bernhard Lederer, Lorenz Oberkofler, Nathan R. Johnson, Fabian Platten.

**Funding acquisition:** Arne Weiberg.

**Investigation:** An-Po Cheng, Lihong Huang.

**Methodology:** An-Po Cheng, Bernhard Lederer, Lorenz Oberkofler, Florian Dunker, Constance Tisserant.

**Project administration:** Arne Weiberg.

**Supervision:** Arne Weiberg.

**Writing – original draft:** An-Po Cheng, Arne Weiberg.

**Writing – review & editing:** An-Po Cheng, Bernhard Lederer, Lorenz Oberkofler, Lihong Huang, Nathan R. Johnson, Fabian Platten, Florian Dunker, Constance Tisserant, Arne Weiberg.

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
