## [Decision Letter · Decision Letter 0]

15 Aug 2023

Dear Dr. Weiberg,

Thank you very much for submitting your manuscript "A fungal RNA-dependent RNA polymerase is a novel player in plant infection and cross-kingdom RNA interference" for consideration at PLOS Pathogens. As with all papers reviewed by the journal, your manuscript was reviewed by members of the editorial board and by several independent reviewers. In light of the reviews (below this email), we would like to invite the resubmission of a significantly-revised version that takes into account the reviewers' comments.

The reviewers appreciated the nice study on an intriguing aspect of plant-pathogen interactions. While they generally appreciated the novel findings presented in the study, they also raised a number of important points.

The reviewers appreciate the elegant use of the “switch-on” GFP reporter system. However, they also ask for additional validation and controls for this part of the study. In the revised version of the manuscript the authors should make sure that these important results are convincingly generated and presented.

The authors should justify the focus on RDR1. As pointed out in the reviews, RDR2 or RDR3 could likewise have a functional relevance in virulence. Hereby, the authors should also interpret their data with respect to previously published results from Weinberg et al, 2013 and discuss the role of RDR1 in the generation of sRNAs involved in ckRNAi.

The authors conclude that reduced virulence of the RDR1 knock-out mutant implies a functional relevance of cross-kingdom RNAi in virulence. As correctly pointed out by the reviewers, the reduced virulence could potentially be caused by other functions related to small RNAs. The authors should clarify their argumentation and more precisely address the role of RDR1.

The study does not include a complementation test of the Botrytis RDR1 mutant. As pointed out by a reviewer, complementation of the rdr1 mutant should be done to validate the functional relevance in ck-RNAi.

One concern also relates the to the small RNA sequencing. From the figures and text it is unclear if the small RNA sequence data is phased or not. This has importance for the interpretation of the data and should therefore be clarified.

In a revised manuscript, the authors should respond to all comments raised by the three reviewers.

We cannot make any decision about publication until we have seen the revised manuscript and your response to the reviewers' comments. Your revised manuscript is also likely to be sent to reviewers for further evaluation.

Sincerely,

Eva H. Stukenbrock, PhD

Academic Editor

PLOS Pathogens

Bart Thomma

Section Editor

PLOS Pathogens

Kasturi Haldar

Editor-in-Chief

PLOS Pathogens

orcid.org/0000-0001-5065-158X

Michael Malim

Editor-in-Chief

PLOS Pathogens

orcid.org/0000-0002-7699-2064

The reviewers appreciated the nice study on an intriguing aspect of plant-pathogen interactions. While they generally appreciated the novel findings presented in the study, they also raised a number of important points.

The reviewers appreciate the elegant use of the “switch-on” GFP reporter system. However, they also ask for additional validation and controls for this part of the study. In the revised version of the manuscript the authors should make sure that these important results are convincingly generated and presented.

The authors should justify the focus on RDR1. As pointed out in the reviews, RDR2 or RDR3 could likewise have a functional relevance in virulence. Hereby, the authors should also interpret their data with respect to previously published results from Weinberg et al, 2013 and discuss the role of RDR1 in the generation of sRNAs involved in ckRNAi.

The authors conclude that reduced virulence of the RDR1 knock-out mutant implies a functional relevance of cross-kingdom RNAi in virulence. As correctly pointed out by the reviewers, the reduced virulence could potentially be caused by other functions related to small RNAs. The authors should clarify their argumentation and more precisely address the role of RDR1.

The study does not include a complementation test of the Botrytis RDR1 mutant. As pointed out by a reviewer, complementation of the rdr1 mutant should be done to validate the functional relevance in ck-RNAi.

One concern also relates the to the small RNA sequencing. From the figures and text it is unclear if the small RNA sequence data is phased or not. This has importance for the interpretation of the data and should therefore be clarified.

In a revised manuscript, the authors should respond to all comments raised by the three reviewers.

Reviewer's Responses to Questions

**Part I - Summary**

Reviewer #1: In this work, the authors present evidence for the role of a Botrytis cinerea RDR protein (referred to as BcRdR1) in achieving cross kingdom (ck)RNAi in both Tomato and Arabidopsis plant models. This conclusion is mainly supported by the genetic deletion of BcRdR1 in two separate lines which results in impaired small RNA accumulation in the pathogen, and defect in pathogenicity. This constitutes the main novelty of this work and a logical continuation to the investigations undertaken by the last author of the study in elucidating the molecular players involved in ckRNAi.

While the evidence that fungal RdR proteins play a role in silencing is of interest, I believe the demonstration could be made stronger with some additional experiments. Furthermore, the work is clearly divided into two sections, one dedicated to exploring the role of BdRdR1 and one dedicated to the generation of in planta tools to a) measure ckRNAi effect in the recipient species using the “switch-on” reporter and b) block the effect of fungal siRNA by titrating them in the recipient host tissue, using the STTM method. While both approaches are ingenious and expand the toolbox to study ckRNAi they have been described in previous paper by the same lab, and do not directly connect to the demonstration of a role of BcRdR1 in sRNA production. The two parts of the paper thus feel disconnected.

Reviewer #2: Introduction:

Cross-kingdom RNA interference (RNAi) is a phenomenon in which small RNA molecules from one organism (usually a pathogen) can enter and function in another organism (usually the host). RNAi is a natural mechanism that regulates gene expression by using small RNA molecules to target and silence specific genes. In the context of cross-kingdom RNAi, the small RNA molecules produced by a pathogen can be transferred into the cells of a host organism, leading to the silencing of host target genes. This process is often exploited by pathogens as a virulence mechanism to promote infection. The transfer of small RNA molecules between different kingdoms of life (e.g., from fungi to plants) through cross-kingdom RNAi is an intriguing and relatively recent discovery. It represents a novel means of communication between organisms and provides a fascinating avenue for understanding the interactions between pathogens and their hosts at the molecular level. Researchers are actively studying cross-kingdom RNAi to gain insights into how pathogens manipulate host gene expression and to explore the potential for using this knowledge to develop new strategies for crop protection and disease control.

Summary:

Small RNAs are the key molecules in cross-kingdom RNA interference, acting as fungal pathogen effectors that silence specific genes in host organisms to facilitate infection. The key factors responsible for producing these small RNAs in pathogens are not well understood. In this study, researchers investigated the role of RNA-dependent RNA polymerase (RDR)1 in pathogenicity and cross-kingdom RNAi in the fungal plant pathogen Botrytis cinerea. They found that RDR1 is essential for both pathogenicity and the production of small RNAs that mediate cross-kingdom RNAi. Using a "switch-on" GFP reporter expressed in living plant tissue, the authors demonstrated that mutants lacking RDR1 showed a reduction in small RNA production, resulting in the loss of cross-kingdom RNA interference and compromised pathogenicity.

Additionally, the study showed that blocking seven pathogen cross-kingdom small RNAs, through the expression of a short-tandem target mimic RNA in transgenic Arabidopsis thaliana, led to reduced infection levels of both the fungal pathogen B. cinerea and the oomycete pathogen Hyaloperonospora arabidopsidis.

In conclusion, these findings highlight the significance of cross-kingdom RNAi in promoting host infection and suggest that pathogen small RNAs could be effective targets for crop protection.

Reviewer #3: The paper by Cheng et al entitled ‘A fungal RNA-dependent RNA polymerase is a novel player in plant infection and cross-kingdom RNA interference’ addresses an important and emerging question in the field that aims to understand the underlying mechanism(s) by which fungal sRNAs are generated that function in ckRNAi. It focuses on the model B. cinerea – Arabidopsis/tomato pathosystem and builds on earlier findings showing Bc-sRNAs are transported into the plant where they hijack the host RNAi machinery to suppress possible defence-related targets. In this paper by Cheng et al the authors focus on the role of BcRDR1 in this ckRNAi. Based on reduced virulence of rdr1 ko mutants in Arabidopsis and tomato and attenuated ckRNAi in ko lines the authors propose that RDR1 is a pathogenicity factor. Although this paper makes exciting findings that link RDRs with ckRNAi there are issues with the manuscript that requires revision before it can be published and as such our initial recommendation would be to invite resubmission with major revision.

**Part II – Major Issues: Key Experiments Required for Acceptance**

Reviewer #1: -The authors detect three putative BcRdR proteins via phylogeny, but focus on RdR1 without providing an explanation. The authors should provide an explanation for focusing on that gene rather than the others. I also think knock out of the two other RdR genes could yield insight into possible redundancy between RdRs in ckRNAi and small RNA biogenesis or lead to a developmental phenotype in the fugus, like it is the case for the double DCL mutant.

-In order to demonstrate the role of BcRdR1, the mutant lines should be complemented by a WT transgenic copy of the gene, which should restore small RNA production. Additionally, a similar construct with a catalytically dead version of the enzyme should be generated, in order to test if production of dsRNA from the Gypsy locus is a prerequisite to BcsRNA production.

-Regarding the small RNA population affected in rdr1 mutant: are reads coming from both strands, as would be expected from sRNA produced from a dsRNA precursor? Since the population of sRNA appears non random from the figure S4A (although it is quite hard to see), are the small RNA phased? Phasing could hint at their biogenesis mode (resulting from a single fungal RISC cleavage/stalled ribosome – downstream recruitment of RdR1?) as seen for plant tasiRNA and LTR retrotransposons.

-in the rdr1 mutant, are the remaining sRNA on gypsy elements still mainly 22nt or 24nt as suggested by figure 2b? This should be made clear in figure S4A (color code legend?).

-Figure 3B is overcropped: there appears to be additional signal under and above the central lane. Please show the uncropped gel. In general, raw data should be made available for all figures.

-Figure 4B is lacking a negative control: t0 of treatment and mock inoculated leaves.

-Figure 4C is the quantification from a single leaf? For a robust conclusion, intensity over several treated leaves should be averaged, and deviation displayed.

Reviewer #2: Strengths and Weaknesses:

In this paper, the authors investigated the role of RDR1 (one of B. cinerea's RNA-dependent RNA polymerases) in pathogenicity and, more importantly, in cross-kingdom RNAi. Their findings may help researchers understand the mechanism of cross-kingdom RNAi. They developed a switch-on GFP reporter, which is a novel technique that can be used to study cross-kingdom RNAi in different species. This technique can also be employed to investigate the role of RNA silencing complexes in both hosts and pathogens, thereby identifying the molecules responsible for RNAi—a significant question in the field of host-microbe interactions.

However, the manuscript needs major improvement, which can be done by additional experiments to confirm their findings. Specifically, to highlight their novel method, the switch-on GFP reporter, it is important to validate their observations using additional methods mentioned below. In the Material and Methods section, the methodology of the experiments is unclear. The authors need to:

1- Clarify the method used to perform inoculation for all infection assays. The authors must provide clear details on how they inoculated the plants for each experiment, including the gene expression assay and switch-on GFP reporter analysis.

2- Specify the reference genes used for mRNA expression or small RNA expression assays. Additionally, explain how they dealt with RNA contamination from either the plant or pathogen while investigating mRNA/small RNA expression. In Figure 1C, they have briefly mentioned the reference gene for measuring biomass, but the authors need to provide detailed information in the Materials and Methods section.

3- Validate some of the microscopy data using other approaches, such as western blotting, especially for the switch-on reporter, which is one of the main techniques used in the experimental procedures and resulted in the key finding.

4- Address the compatibility issues of some results presented in different figures/panels. The authors should either remove conflicting data or provide explanations in the text.

5- Improve the figure legends by including more detail. Authors can add a short explanation of their methodology in the figure legends, particularly for the infection assay, to clarify how the inoculation was performed in each figure or for gene expression assay.

5- Simplify or rewrite some paragraphs that are complicated to understand, making them easier to follow for readers.

Comments and suggestions based on text and figures:

Major:

Line 124 to 126: The authors reported that the mRNA level of BcGypsy3 increases in the Bcrdr1 mutant, suggesting that BcRDR1 may play a role in post-transcriptional silencing of retrotransposons (RTs). However, the authors' confidence in this statement is not explicitly mentioned. It is possible that the accumulation of BcGypsy3 mRNA in the Bcrdr1 mutant is a result of compromised sRNA production in Bcdrd1. In the wild-type B. cinerea, RDR1 is involved in generating dsRNA molecules from single-stranded BcGypsy3 mRNA, which are subsequently targeted by DCLs (Dicer-like proteins), leading to the production of small RNAs (sRNAs). However, in the bcrdr1 mutant, the BcGypsy3 mRNAs are not affected by this process, potentially resulting in the accumulation of BcGypsy3 mRNA. Further investigations are needed to ascertain the precise role of BcRDR1 in the post-transcriptional silencing of RTs and its impact on sRNA production in B. cinerea.

Line 139: I couldn't find Figure 3D mentioned in line 139. Figure 3 only has three panel: A, B, and C. Please clarify which panel belong to 3D.

line 139 to 144 (related to both figure 3C and S5A,B) the authors stated that they measured successful infection by analyzing the expression pattern of S. lycopersicum SlPI-I and SlPI-II or A. thaliana AtPDF1.2 immunity genes. If the infection is successful and B. cinerea possesses effective pathogenicity, it is expected to observe up-regulation of AtPDF1.2 in arabidopsis. However, in Figure 3C and Figure S5A-B, Arabidopsis infected by bcrdr1 is actually showing a higher expression of AtPDF1.2, which indicates that bcrdr1 and bcrdr2 mutants have a higher level of infection, does that mean higher level of pathogenicity? If yes, this finding is inconsistent with the authors' conclusion in line 139 to 144.

Line 309 to 313: This paragraph is unclear. Although the paper contains numerous infection and pathogenicity assays, it is not clear how they performed the infection assay. For instance, in lines 317 to 321, it states:

"For recording GFP cross-kingdom RNAi reporter time course, 3-week-old A. thaliana were cultivated on 1⁄2 MS + 1% sucrose agar plates and inoculated with 2x105 conidia/ml B. cinerea resuspended in 1% malt extract medium pre-incubated for 1 hour. The pre-incubated conidia suspension was washed twice with sterile water before 5 μl was dropped at the center of one leaf per seedling."

It is unclear whether they sprayed conidia over the leaf or only used a droplet at the center. If they used a 5 μl droplet as mentioned in the text, the authors need to explain how 5 μl droplet of conidia in the center of the leaf can cause infection in the entire leaf and induce the reporter in the entire leaf, as presented in Figure 4B and supporting video 1.

Figure 4B: This is one of the key results and requires additional validation through another method, such as western blotting. To confirm the findings, it is recommended to repeat the same experiment using same timepoints and perform a western blot using anti-GFP antibodies to demonstrate that higher expression of GFP only occurs when plants are infected by WT B. cinerea. To ensure that each sample contains the same amount of protein, either Ponceau staining or silver staining can be used.

Figure S2A:

In this figure, the authors aimed to evaluate the expression pattern of RDR1, RDR2, and RDR3 in B. cinerea both in axenic media and during infection. Line 98 to 99: they were unable to detect any significant up-regulation during infection, but they did not provide detailed information about the experiment in method section. One of the main challenges in evaluating gene expression during infection is dealing with the ratio of plant and fungal pathogen RNAs. The main questions raised are: What is their RNA isolation method? How did they quantify plant or fungal RNAs? And what reference genes were used to normalize the amount of plant or fungal RNAs?

Figure S6: Similar to what has been suggested for the result in Figure 4B, here I also recommend using western blotting (WB) and anti-GFP antibodies to validate your result. This can be achieved by using WT B. cinerea, bcrdr1#2, and bcdcl1/2 at two time points: 6 hours and 24 hours post-infection.

Reviewer #3: 1: Through “genome survey” the authors identify 3 BcRDRs in Bc and show in Suppl Fig 2 that all three genes are transcribed when the fungus is grown axenically. They then go on to generate a mutant in only one of the isoforms, BcRDR1 (homolog of NcSAD1 involved in MSUD). Why not RDR2 or RDR3 as well? I understand that RDR3 does not have homology to any well-characterised RDRs in fungi so exploring BcRDR3 role might be a bit of a dead end (although its downregulation during infection might be very interesting), but given that RDR2 is an orthologue of NcQDE1, might it not be worth exploring too?

- Given the jump from the above paragraph Line 88-102, where all three BcRDRs are discussed, to just focussing on RDR1 in future experiments, I feel that some justification for the sole focus on RDR1 is required especially since all three RDRs contain the conserved motif needed for RNA polymerase function.

2. Using bcrdr1 knockout (ko) mutants the authors show mutants have reduced lesion size and fungal biomass on Arabidopsis and tomato detached leaves while axenically grown ko mutants appear comparable to WT (Fig 1A-D and Fig S3). In Lines 107-108 based on these results the authors conclude that BcRDR1 is a pathogenicity factor required to generate sRNAs needed for ckRNAi. Surely a reduction in the ability of Bc to cause disease, ie the virulence of ko Bc strains would also produce a reduction in lesion size during infection which may be unrelated to ckRNAi? For example, in Bc ko lines a compromise in necrosis inducing ability (e.g. failure to induce CWDE in ko lines) would also result in reduced lesion size and consequently a reduction in the production of Bc-sRNAs so results presented could be unrelated to ckRNAi. A more in-depth characterisation of fungal virulence (rather than just measuring axenic mycelial growth and BcTUB expression in infected tissue) should be provided.

3. In Fig S2 the authors show expression of all 3 RDRs are suppressed during infection. However, in an earlier publication (Weiberg et al 2013) the authors show candidate Bc-siRNAs from this study are significantly upregulated during infection. Therefore, it seems counterintuitive/unlikely that RDR1 is required for the generation of Bc-sRNAs involved in ckRNAi during infection. Please could the authors clarify/comment on this?

4. Attenuated expression of Bc-sRNAs is shown in ko lines grown axenically. However, based on Weiberg et al 2013 showing Bc-sRNAs upregulated during infection to confirm if Bc-sRNAs are RDR1-dependent could the authors please confirm that Bc-sRNA expression is compromised in bcrdr1 ko lines in infected tissue (normalised to fungal biomass)?

5. In line 124-126 the authors show that “accumulation of RT-derived Bc-sRNAs in bcrdr1 ko mutants resulted in increased mRNA levels of BcGypsy3, (Fig S4), suggesting a role of BcRDR1 also in post-transcriptional silencing of RTs.” FigS4 shows both BcGypsy1 and 3 are upregulated in ko line 4.

- Could the authors show RDR1-dependent sRNAs that would target BcGypsy1/3?

- What would the consequence be of upregulated BcGypsy1/3 on the fungus especially ito its virulence?

4. The authors generate a ‘switch-on’ reporter to monitor ckRNAi and focus on two sRNA targets for Bc-sRNAs (siRNA3.1 and 3.2). Could the authors explain why they chose these two siRNAs (which are generated from the same locus) over siRNA5 and siRNA20?

- As a matter of interest, could the authors clarify why both target sites were included in the reporter rather than testing Bc-sRNA target sites individually (perhaps in duplicate) in the reporter? With both included it remains unclear if only one or both are targeted by Bc-sRNAs. Please clarify?

- To demonstrate specificity for target sequences scrambled control target sequence should be included or a control Bc sRNA target that is not modulated through ckRNAi. This data should be included.

- In Line 152-153 “Csy4 constantly suppresses expression of the GFP, unless GFP expression is activated when Bc-sRNAs silence Csy4.” However, Fig 4D shows GFP expression in WT during infection comparable to that of non-infected plants. Indeed, only one datapoint in WT shows slightly elevated GFP transcript levels. As such this data is not convincing as excluding the single datapoint would show no difference in GFP expression in the presence or absence of Bc.

- Similarly, the WB in Fig 4E also shows presence of GFP in reporter lines in the absence of the fungus. This suggests a leakiness in the reporter system which could be due to Bc-sRNA target sites being targeted by endogenous plant sRNAs. Could this be due to endogenous host sRNAs with potential to target the same Bc-sRNA3.1/3.2 targets and that could switch on the reporter in colonised tissue?

- Also, could reduced levels of GFP in ko lines be due to reduced levels of endogenous host sRNAs in response to lower fungal biomass? Is 10x more conidiospores mentioned in Line 172 sufficiently high especially since BcTUB is still shown to be low?

- Perhaps reporter leakiness is due to promoter (proEF) choice?

- Fig 7S shows ko line 4 also being more ‘leaky’ than line 2; please could the authors comment on this?

The authors should address these comments and include experiments/controls to validate the reporter; since with GFP ON in the absence of fungal infection, the role of RDR1 in ckRNAi remains inconclusive.

5. Figure 4B shows a time series of GFP expression in Bc infected WT reporter lines compared to the bcrdr1 line 2 and in Fig 4C GFP intensity is quantified. Could the authors please include a control that lacks GFP to show that the fluorescence observed is not due to autofluorescence from colonised tissue. Similarly, could the authors stain fungal hyphae with WGA-Alexafluor or Trypan Blue to confirm if GFP reporter activation coincide with the fungal colonisation – especially at the infection front as shown in Fig S7B.

**Part III – Minor Issues: Editorial and Data Presentation Modifications**

Reviewer #1: -Why are small RNA sequenced from an axenic culture rather than during plant infection? It seems intuitive that if BcsRNA have a pathogenic role, they would be induced during infection.

-Similarly, it is counterintuitive that BcRdR1 decreases during infection (figure S2A). This should be commented on by the authors.

-definition of “multiple mapping reads” is unclear: do the authors mean that they allowed mapping of single reads to multiple loci as an approximation of sRNA made from repeat elements?

-For the switch-on reporter, why were sites for siR3.2 and siR3.1 specifically picked for this assay? And for STTM assay, why siR3.2 and siR5? The choice of those over other described sRNA target sites is never made clear

-There is a discrepancy between microscopy images presented in figure 4B and figure S6 and S7. The former shows accumulation of GFP in the midrib and veins, which is still visible in presence or bcrdr1 ko, while the former doesn’t show veins at al. This and the point above make it impossible to know if the leftover GFP signal is the result of incomplete Csy4 activity/insufficient expression, or of residual BdsiRNA in bdrdr1.

-Figure S7 legend is very unclear: refers to a scale bar in C, which is apparently a rtQPCR assay, which is never made clear in legend text. Same for panel E.

-Figure S7E: there is a difference in BcTUB accumulation between WT and reporter containing Arabidopsis plants. This suggests there is a difference on the pathogenicity of the fugus imparted by the construct, unless different amounts were inoculated.

-line 116-117: in depth?

-line 130 : Vacuolar to replace Valuolar

Reviewer #2: Minor:

Line 136 to 139: This sentence is complicated and difficult to follow. It needs to be divided into simple sentences.

“When infecting S. lycopersicum or A. thaliana with B. cinerea bcrdr1 ko mutants, down-regulation of Bc-sRNA target genes, as observed in WT-infected plants infected with the wild-type strain compared to non-inoculated plants, was abolished or less strong (Figure 3C-D, Figure S5A-B).”

Figure 3B:

Please state which Tubulin gene was used as the reference gene, α-TUB or β-TUB? And for future reference, please note that UBQ appears to show more stable expression than α-TUB/β-TUB as a reference gene in B. cinerea.

Please see: Ren H, Wu X, Lyu Y, Zhou H, Xie X, Zhang X, Yang H. Selection of reliable reference genes for gene expression studies in Botrytis cinerea. J Microbiol Methods. 2017 Nov;142:71-75. doi: 10.1016/j.mimet.2017.09.006. Epub 2017 Sep 14. PMID: 28917607.

Table S1: The first sheet name appears to be in another language, possibly Chinese. Please change it to English.

Figure S6 panel 1: It appears that for brdcl1/2, the GFP panel and bright field (BF) panel are from different imaging. Although there is no GFP signal, you should be able to see the same cell pattern in both the GFP and BF images when changing the contrast of the images. Please carefully review all microscopy images.

Reviewer #3: Lines 38-39: “The cellular-type RDRs are conserved in several eukaryote kingdoms, including plants, nematodes, and fungi”

- If using the term “kingdoms”, nematodes should be listed under animals – if wanting to make the point that RDRs are found in some animals like nematodes, but not in e.g. mammals, this should be reworded

Line 88: SPAG- “we” not “were”

Line 89: “Through a genome survey”

- What does genome survey mean in this context? Genome survey often refers to genomic structure which I assume the authors are not referring to, thus how were the three RDR-encoding genes identified - by homology to characterised RDRs (e.g. SAD-1/QDE-1), gene annotation searching, protein domain analysis? This specific approach used should be clarified here, and should also be described in Materials and Methods.

- On a related note, I would advise the authors to include greater detail with respect to any in silico analyses (both this and phylogenetic analysis) in the supplementary materials. While gene loci identifiers are provided throughout, a table describing the source of the sequence data used (e.g. NCBI/Phytozome/ensemblPlants) and the date this resource was accessed will greatly improve replicatibility of the study - as would the transcript IDs used in phylogenetic analysis.

Lines 98-99: “however, there was no significant up-regulation of any of the BcRDRs measured during infection (Figure S2A).”

- In fact, there appears to be significant downregulation of RDR1 at 2dpi, and RDR3 from 1dpi onwards- please could the authors comment on/ discuss the implications thereof?

Lines 104-105: “lower fungal biomass, as estimated by quantification of B. cinerea Tubulin mRNAs levels”

- The authors later state axenic growth is not impaired in the mutants but it would be reassuring to see that BcTubulin expression is also not affected in the knockout mutants relative to wild type strains to provide additional direct evidence for the authors’ conclusion. Indeed, this would be particularly useful to show in relation to Figure S7E which shows reduced BcTubulin expression in mutants even with 10x greater conidia concentrations used for inoculation.

Lines 107-108: “Based on these results, we assume that BcRDR1 is a pathogenicity factor in B. cinerea”

- Careful of wording here- “assume” implies a judgement without evidence- better to say “we believe”, or “BcRDR1 is likely a pathogenicity factor in B.cinerea”

Line 113: “B. cinerea Bc-sRNA biogenesis”

- Redundant- either “B.cinerea sRNA biogenesis” or “Bc-sRNA biogenesis”

Line 130: SPAG- “Vacuolar” not “valuolar”

Line 133: “and the A. thaliana host genes AtMPK1 (AT1G10210), AtMPK2”

- All other genes are given full names here, these should be too

Line 134: “Peroxiredoxin” not “Peroxredoxin”

Lines 141-142: “S. lycopersicum Sl Proteinase inhibitor (PI)-I and SlPI-II or A. thaliana AtPlant Defensin (PDF)1.2 immunity genes”

- Redundancy- can remove the “Sl” and “At”

Paragraph 145- 176

- Is it correct to claim that this is a “novel” reporter system for ckRNAi- to my knowledge this has been done before in the Dunker et al. 2020 paper- I understand that the GUS has been switched out for GFP in this case but does this really add this much novelty, as it is still operating on the same principles

- To be accurate this should be rephrased to better reflect the adaptation of this ckRNAi reporter and the Dunker 2020 paper should be cited as the first instance of using this Csy4-based ckRNAi reporter– also, why was the decision made to switch out GUS for GFP? This does not necessarily need to be explained in the main text but would be interesting to know.

- Lines 174-176 refer to observations that strongest GFP activity was identified at the infection front after inoculation with WT B. cinerea but, while interesting, it is not immediately apparent how this relates to the overarching results that bcrdr1 mutants are compromised in ckRNAi - was this observation not found upon inoculation with the mutant strains?

- How did the authors discriminate between increased GFP expression in infected material and increased autofluorescence due to higher fungal infection compared to ko lines. Please could the authors provide controls to discriminate between GFP signal and autofluorescence.

Lines 191-193: “Bc-sRNAs STTM plants exhibited reduced lesion sizes induced by B. cinerea, compared to A. thaliana WT or a transgenic A. thaliana line expressing a non-sense RNA-STTM”

- I am assuming that the authors mean a “nonsense” STTM- i.e. one with a scrambled sequence that will not lock the sRNAs- this is made a bit unclear with the hyphenation of “nonsense” to “non-sense” which to me implies something related to sense and antisense strands- I think this should be changed to make the meaning clearer (again in line 195 and in figure 5B, 5C, S8)

Lines 210-211: “A. thaliana master STTM plants grew and developed normally and did not show any pleiotropic defect.” Please could the author show the data?

Lines 215-216: “Reduced infection was not due to constantly enhanced plant immunity”

- The authors’ conclusion is drawn from evidence that Pseudomonas syringae infection is not affected in the master STTM plants. While this is useful evidence, this is assuming fungal, oomycete and bacterial pathogens all elicit the same immune response. Better evidence for this claim would be to show that the expression of plant immunity-associated genes (other than those targeted by the pathogens) in the master STTM plants is not affected upon inoculation with the assayed pathogens.

Lines 217-218: “did not result in fewer reduced bacterial colonies” -> either fewer or reduced, not both

Lines 230-232: The authors mention the bcdcl1dcl2 ko in the ku70 background that showed no evidence for ckRNAi – what point is the authors making here?

Lines 268-270: “It will be interesting to investigate whether microbial RDRs play a broader role in diverse plant-microbe interactions and in cross-kingdom RNA communication.”

- As the targeting of BcDCLs using SIGS has already been mentioned in this paragraph, it might be nice to further extend this conclusion to suggest how this research could be extended to have some practical applications i.e. the identification of a new fungal pathogenicity factor (BcRDR1) that could similarly be targeted using SIGS in disease control

Line 325: “using a binocular.” -> “using a binocular microscope.”

Line 378: “by default multiple alignment algorithms.”

- While the authors have included the software and version number used in this analysis, this software is proprietary and all readers may not have access to it. It would be nice to at least include what these default settings are, even if just as “... algorithms (gap open cost: X; gap extension cost: Y).” to enable replication using alternative software.

Line 505: “The scale bars in A) and C) represent 500 μm”

- Should this be “A) and B)”?

Figure S1:

- This figure would benefit from an explanation of the ‘Max’ and ‘Min’ colour spectrum used within (whether in the caption or by adjusting these labels). Does this refer to residue conservation or another measure of alignment/homology?

Figures S6-S7

- The brightfield images should be included as they were taken (i.e. before merging with GFP) such that the figures show BF, GFP, merged.

PLOS authors have the option to publish the peer review history of their article (what does this mean?). If published, this will include your full peer review and any attached files.

Reviewer #1: No

Reviewer #2: No

Reviewer #3: No
---

## [Decision Letter · Decision Letter 1]

10 Nov 2023

Dear Dr. Weiberg,

Thank you very much for submitting your manuscript "A fungal RNA-dependent RNA polymerase is a novel player in plant infection and cross-kingdom RNA interference" for consideration at PLOS Pathogens. As with all papers reviewed by the journal, your manuscript was reviewed by members of the editorial board and by several independent reviewers. The reviewers appreciated the attention to an important topic. Based on the reviews, we are likely to accept this manuscript for publication, providing that you modify the manuscript according to the review recommendations.

The three reviewers appreciated the careful consideration of all points raised in the previous review, and they found the revised manuscript -mostly - of excellent quality. However, a few important points were raised regarding interpretation and discussion of notably the small RNA sequencing. Therefore, the authors should carefully consider the raised points and amend their manuscript with a more detailed discussion of their results.

Sincerely,

Eva H. Stukenbrock, PhD

Academic Editor

PLOS Pathogens

Bart Thomma

Section Editor

PLOS Pathogens

Kasturi Haldar

Editor-in-Chief

PLOS Pathogens

orcid.org/0000-0001-5065-158X

Michael Malim

Editor-in-Chief

PLOS Pathogens

orcid.org/0000-0002-7699-2064

The three reviewers appreciated the careful consideration of all points raised in the previous review, and they found the revised manuscript -mostly - of excellent quality. However, a few points important points were raised regarding interpretation and discussion of notably the small RNA sequencing. Therefore, the authors should carefully consider the raised points and amend their manuscript with a more detailed discussion of their results.

Reviewer Comments (if any, and for reference):

Reviewer's Responses to Questions

**Part I - Summary**

Reviewer #1: Second Review of Manuscript “A fungal RNA-dependent RNA polymerase is a novel player in plant infection and cross-kingdom RNA interference” by An-Po Cheng & colleagues.

First, I would like to thank the authors for thoroughly addressing most raised points, and to state that I am satisfied with the revisions as they stand, with one main exception: the reanalysis of siRNA from BcGypsy loci. The results of the small RNA sequencing are rather curious and suggest that BcRDR1 and double stranded RNA production is not directly responsible for siRNA produced from BcGypsy3 and 1. There are however a few things left to discuss from the data added by the authors:

Reviewer #2: The authors have addressed all questions and provided additional information regarding their materials and methods. They have also included additional data to support the switch-on GFP reporter, which was necessary as this technique was first reported by them and needed validation. The minor revisions have been addressed, and the manuscript is now in excellent structural and technical shape, making it an interesting piece of science with clear and followable materials and methods. It's also an enjoyable read for the interested audience. Therefore, no further revisions are needed.

Reviewer #3: All concerns raised in our previous review have been adequately and sufficiently addressed by the authors. We recommend this article for publication without further revision.

**Part II – Major Issues: Key Experiments Required for Acceptance**

Reviewer #1: -In Figure R1 the authors show that BcGypsy1 and 3 siRNAs do not result from phased biogenesis unlike tasiRNA of plant. The maximum RPM for 22-nt BcGypsy3 is just over 30.000, but in figure sup3B rep1 goes up to 120.000. how come?

-in Figure R1, where are the atTAS1A reads sampled from? Is the data borrowed from another paper or from the authors own datasets? If so, this cannot be from axenic culture as in the data included in this study.

-In Figure S7A, it appears that most siRNA from BcGypsy3 are coming from the negative (-) strand? This is mostly consistent with Porquier et al. 2021, where most siRNA are also produced from the - strand, in axenic and infected samples. However, a fair number of reads are also coming from the + strand in that previous study. Can the authors explain the discrepancy with the current study? I also noticed a significant difference in 5’ bias in this study with Porquier et al. 2021 (compare figure 2C here with figure 2D in previous paper). The bias is much less pronounced here. Could this reflect a difference in library preparation between the two studies? A difference in loading and stability could influence apparent abundance of siRNA and if a given strand is fed into AGO-complexes, raise the abundance of siRNA coming from this strand over the other and could still be compatible with direct dsRNA production from BcRDR1.

In any case, this surprising result should be discussed at the end of the study and deserves its own paragraph.

Reviewer #2: (No Response)

Reviewer #3: N/A

**Part III – Minor Issues: Editorial and Data Presentation Modifications**

Reviewer #1: -In supS7D: upregulation only in WT?

-Figure S7A: orientation of the transcription should be indicated with an arrow as well as the different features on the gypsy element.

-Figure 1A, 1B and 1C should be called in the text appropriately.

-Line 125-126 and 129: Why past tense?

-Line 135-136: In total, 1.3 – 4.9 million reads were mapped to the B. cinerea reference genome either unique or multiple times, as to the same read mapped to multiple chromosomal regions. Unclear sentence

Reviewer #2: (No Response)

Reviewer #3: N/A

PLOS authors have the option to publish the peer review history of their article (what does this mean?). If published, this will include your full peer review and any attached files.

Reviewer #1: No

Reviewer #2: No

Reviewer #3: No

Figure Files:

Data Requirements:

Reproducibility:

References:

---

## [Editor Report · Decision Letter 2]

5 Dec 2023

Dear Dr. Weiberg,

We are pleased to inform you that your manuscript 'A fungal RNA-dependent RNA polymerase is a novel player in plant infection and cross-kingdom RNA interference' has been provisionally accepted for publication in PLOS Pathogens.

Best regards,

Eva H. Stukenbrock, PhD

Academic Editor

PLOS Pathogens

Bart Thomma

Section Editor

PLOS Pathogens

Kasturi Haldar

Editor-in-Chief

PLOS Pathogens

orcid.org/0000-0001-5065-158X

Michael Malim

Editor-in-Chief

PLOS Pathogens

orcid.org/0000-0002-7699-2064
---

## [Editor Report · Acceptance letter]

14 Dec 2023

Dear Dr. Weiberg,

We are delighted to inform you that your manuscript, "A fungal RNA-dependent RNA polymerase is a novel player in plant infection and cross-kingdom RNA interference," has been formally accepted for publication in PLOS Pathogens.

Best regards,

Michael Malim

Editor-in-Chief

PLOS Pathogens

orcid.org/0000-0002-7699-2064